# Fast and Complete: Enabling Complete Neural Network Verification with Rapid and Massively Parallel Incomplete Verifiers

**Kaidi Xu**[*,1]     **Huan Zhang**[*,2]     **Shiqi Wang**[3]     **Yihan Wang**[2]
**Suman Jana**[3]     **Xue Lin**[1]     **Cho-Jui Hsieh**[2]

[1]Northeastern University   [2]UCLA   [3]Columbia University

`xu.kaid@northeastern.edu, huan@huan-zhang.com, tcwangshiqi@cs.columbia.edu,`
`wangyihan617@gmail.com, suman@cs.columbia.edu, xue.lin@northeastern.edu,`
`chohsieh@cs.ucla.edu`

## Abstract

Formal verification of neural networks (NNs) is a challenging and important problem. Existing efficient complete solvers typically require the branch-and-bound (BaB) process, which splits the problem domain into sub-domains and solves each sub-domain using faster but weaker incomplete verifiers, such as Linear Programming (LP) on linearly relaxed sub-domains. In this paper, we propose to use the backward mode linear relaxation based perturbation analysis (LiRPA) to replace LP during the BaB process, which can be efficiently implemented on the typical machine learning accelerators such as GPUs and TPUs. However, unlike LP, LiRPA when applied naively can produce much weaker bounds and even cannot check certain conflicts of sub-domains during splitting, making the entire procedure incomplete after BaB. To address these challenges, we apply a fast gradient based bound tightening procedure combined with batch splits and the design of minimal usage of LP bound procedure, enabling us to effectively use LiRPA on the accelerator hardware for the challenging complete NN verification problem and significantly outperform LP-based approaches. On a single GPU, we demonstrate an order of magnitude speedup compared to existing LP-based approaches.

## 1 Introduction

Although neural networks (NNs) have achieved great success on various complicated tasks, they remain susceptible to adversarial examples (Szegedy et al., 2013): imperceptible perturbations of test samples might unexpectedly change the NN predictions. Therefore, it is crucial to conduct formal verification for NNs such that they can be adopted in safety or security-critical settings.

Formally, the neural network verification problem can be cast into the following decision problem:

*Given a neural network $f(\cdot)$, an input domain $\mathcal{C}$, and a property $\mathcal{P}$. $\forall x \in \mathcal{C}$, does $f(x)$ satisfy $\mathcal{P}$?*

The property $\mathcal{P}$ is typically a set of desirable outputs of the NN conditioned on the inputs. Typically, consider a binary classifier $f(x)$ and a positive example $x_0$ ($f(x_0) \geq 0$), we can set $\mathcal{P}$ to be nonnegative numbers $\mathbb{R}^+$ and $x$ is bounded within an $l_\infty$ norm ball $\mathcal{C} = \{x | \|x - x_0\|_\infty \leq \epsilon\}$. The success of verification guarantees that the label of $x_0$ cannot flip for any perturbed inputs within $\mathcal{C}$.

In this paper we study the *complete verification* setting, where given sufficient time, the verifier should give a definite "yes/no" answer for a property under verification. In the above setting, it must solve the non-convex optimization problem $\min_{x \in \mathcal{C}} f(x)$ to a global minimum. Complete NN verification is generally a challenging NP-Hard problem (Katz et al., 2017) which usually requires expensive formal verification methods such as SMT (Katz et al., 2017) or MILP solvers (Tjeng et al., 2019b). On the other hand, incomplete solvers such as convex relaxations of NNs (Salman et al., 2019) can only provide a sound analysis, i.e., they can only approximate the lower bound of $\min_{x \in \mathcal{C}} f(x)$ as $\underline{f}$ and verify the property when $\underline{f} \geq 0$. No conclusion can be drawn when $\underline{f} < 0$.

Recently, a Branch and Bound (BaB) style framework (Bunel et al., 2018; 2020b) has been adopted for efficient complete verification. BaB solves the optimization problem $\min_{x \in \mathcal{C}} f(x)$ to a global

minimum by branching into multiple sub-domains recursively and bounding the solution for each sub-domain using incomplete verifiers. BaB typically uses a Linear Programming (LP) bounding procedure as an incomplete verifier to provide feasibility checking and relatively tight bounds for each sub-domain. However, the relatively high solving cost of LPs and incapability of parallelization (especially on massively parallel hardware accelerators like GPUs or TPUs) greatly limit the performance and scalability of the existing complete BaB based verifiers.

In this paper, we aim to use fast and typically weak incomplete verifiers for complete verification. Specifically, we focus on a class of incomplete verifiers using efficient bound propagation operations, referred to as linear relaxation based perturbation analysis (LiRPA) algorithms (Xu et al., 2020). Representative algorithms in this class include convex outer adversarial polytope (Wong & Kolter, 2018), CROWN (Zhang et al., 2018) and DeepPoly (Singh et al., 2019b). LiRPA algorithms exhibit high parallelism as the bound propagation process is similar to forward or backward propagation of NNs, which can fully exploit machine learning accelerators (e.g., GPUs and TPUs).

Although LiRPA bounds are very efficient for incomplete verification, especially in training certified adversarial defenses (Wong et al., 2018; Mirman et al., 2018; Wang et al., 2018a; Zhang et al., 2020), they are generally considered too loose to be useful compared to LPs in the complete verification settings with BaB. As we will demonstrate later, using LiRPA bounds naively in BaB cannot even guarantee the completeness when splitting ReLU nodes, and thus we need additional measures to make them useful for complete verification. In fact, LiRPA methods have been used to get upper and lower bounds for each ReLU neuron in constructing tighter LPs (Bunel et al., 2018; Lu & Kumar, 2020). It was also used in (Wang et al., 2018c) for verifying small-scale problems with relatively low dimensional input domains using input splits, but splitting the input space can be quite ineffective (Bunel et al., 2018) and is unable to scale to high dimensional input case like CIFAR-10. Except one concurrent work (Bunel et al., 2020a), most complete verifiers are based on relatively expensive solvers like LP and cannot fully take benefit from massively parallel hardware (e.g., GPUs) to obtain tight bounds for accelerating large-scale complete verification problems. Our main contributions are:

• We show that LiRPA bounds, when improved with fast gradient optimizers, can potentially outperform bounds obtained by LP verifiers. This is because LiRPA allows joint optimization of both intermediate layer bounds of ReLU neurons (which determine the tightness of relaxation) and output bounds, while LP can only optimize output bounds with fixed relaxations on ReLU neurons.

• We show that BaB purely using LiRPA bounds is insufficient for complete verification due to the lack of feasibility checking for ReLU node splits. To address this issue, we design our algorithm to only invoke LP when absolutely necessary and exploits hardware parallelism when possible.

• To fully exploit the hardware parallelism on the machine learning accelerators, we use a batch splitting approach that splits on multiple neurons simultaneously, further improving our efficiency.

• On a few standard and representative benchmarks, our proposed NN verification framework can outperform previous baselines significantly, with a speedup of around 30X compared to basic BaB+LP baselines, and up to 3X compared to recent state-of-the-art complete verifiers.

## 2 BACKGROUND

### 2.1 FORMAL DEFINITION OF NEURAL NETWORK (NN) VERIFICATION

**Notations of NN.** For illustration, we define an $L$-layer feedforward NN $f : \mathbb{R}^{|x|} \to \mathbb{R}$ with $L$ weights $\mathbf{W}^{(i)}$ ($i \in \{1, \cdots, L\}$) recursively as $h^{(i)}(x) = \mathbf{W}^{(i)} g^{(i-1)}(x)$, hidden layer $g^{(i)}(x) = \text{ReLU}(h^{(i)}(x))$, input layer $g^{(0)}(x) = x$, and final output $f(x) = h^{(L)}(x)$. For simplicity we ignore biases. We sometimes omit $x$ and use $h_j^{(i)}$ to represent the *pre-activation* of the $j$-th ReLU neuron in $i$-th layer for $x \in \mathcal{C}$, and we use $g_j^{(i)}$ to represent the post-activation value. We focus on verifying ReLU based NNs, but our method is generalizable to other activation functions supported by LiRPA.

**NN Verification Problem.** Given an input $x$, its bounded input domain $\mathcal{C}$, and a feedforward NN $f(\cdot)$, the aim of formal verification is to prove or disprove certain properties $\mathcal{P}$ of NN outputs. Since most properties studied in previous works can be expressed as a Boolean expression over a linear equation of network output, where the linear property can be merged into the last layer weights of a NN, the ultimate goal of complete verification reduces to prove or disprove:

$$\forall x \in \mathcal{C}, f(x) \geq 0 \tag{1}$$

One way to prove Eq. 1 is to solve $\min_{x \in \mathcal{C}} f(x)$. Due to the non-convexity of NNs, finding the exact minimum of $f(x)$ over $x \in \mathcal{C}$ is challenging as the optimization process is generally NP-complete (Katz et al., 2017). However, in practice, a *sound* approximation of the lower bound for $f(x)$, denoted as $\underline{f}$, can be more easily obtained and is sufficient to verify the property. Thus, a good verification strategy to get a tight approximation $\underline{f}$ can save significant time cost. Note that $\underline{f}$ must be sound, i.e., $\forall x \in \mathcal{C}, \underline{f} \leq f(x)$, proving $\underline{f} \geq 0$ is sufficient to prove the property $f(x) \geq 0$.

## 2.2 THE BRANCH AND BOUND (BaB) FRAMEWORK FOR NEURAL NETWORK VERIFICATION

Branch and Bound (BaB), an effective strategy in solving traditional combinatorial optimization problems, has been customized and widely adopted for NN verification (Bunel et al., 2018; 2020b). Specifically, BaB based verification framework is a recursive process, consisting of two main steps: branching and bounding. For *branching*, BaB based methods will divide the bounded input domain $\mathcal{C}$ into sub-domains $\{\mathcal{C}_i | \mathcal{C} = \bigcup_i \mathcal{C}_i\}$, each defined as a new independent verification problem. For instance, it can split a ReLU unit $g_j^{(k)} = \text{ReLU}(h_j^{(k)})$ to be negative and positive cases as $\mathcal{C}_0 = \mathcal{C} \cap \left( h_j^{(k)} \geq 0 \right)$ and $\mathcal{C}_1 = \mathcal{C} \cap \left( h_j^{(k)} < 0 \right)$ for a ReLU-based network; for each sub-domain $\mathcal{C}_i$, BaB based methods perform *bounding* to obtain a relaxed but sound lower bound $\underline{f}_{\mathcal{C}_i}$. A tightened global lower bound over $\mathcal{C}$ can then be obtained by taking the minimum values of the sub-domain lower bounds from all the sub-domains: $\underline{f} = \min_i \underline{f}_{\mathcal{C}_i}$. Branching and bounding will be performed recursively to tighten the approximated global lower bound over $\mathcal{C}$ until either (1) the global lower bound $\underline{f}$ becomes larger than 0 and prove the property or (2) a violation (e.g., adversarial example) is located in a sub-domain to disprove the property. Essentially, we build a search tree where each leaf is a sub-domain, and the property $\mathcal{P}$ can be proven only when it is valid on all leaves.

**Soundness of BaB** We say the verification process is sound if we can always trust the "yes" ($\mathcal{P}$ is verified) answer given by the verifier. It is straightforward to see that the whole BaB based verification process is sound as long as the bounding method used for each sub-domain $\mathcal{C}_i$ is sound.

**Completeness of BaB** The completeness of the BaB-based NN verification process, which was usually assumed true in some previous works (Bunel et al., 2020b; 2018), in fact, is not always true even if all possible sub-domains are considered with a sound bounding method. Additional requirements for the bounding method are required - we point out that a key factor for completeness involves feasibility checking in the bounding method which we will discuss in Section 3.2.

**Branching in BaB** Since branching step determines the shape of the search tree, the main challenge is to efficiently choose a good leaf to split, which can significantly reduce the total number of branches and running time. In this work we focus on branching on activation (ReLU) nodes. BaBSR (Bunel et al., 2018) includes a simple branching heuristic which assigns each ReLU node a score to estimate the improvement for tightening $\underline{f}$ by splitting it, and splits the node with the highest score.

**Bounding with Linear Programming (LP)** A typical bounding method used in BaB based verification is the *Linear Programming bounding procedure* (sometimes simply referred to as "LP" or "LP verifier" in our paper). Specifically, we transform the original verification problem into a linear programming problem by relaxing every activation unit as a convex (linear) domain (Ehlers, 2017) and then get the lower bound $\underline{f}_{\mathcal{C}_i}$ with a linear solver given domain $\mathcal{C}_i$. For instance, as shown in Figure 1a, $g_j^{(i)} = \text{ReLU}(h_j^{(i)})$ can be linearly relaxed with the following 3 constraints: (1) $g_j^{(i)} \geq h_j^{(i)}$; (2) $g_j^{(i)} \geq 0$; (3) $g_j^{(i)} \leq \frac{\boldsymbol{u}_j^{(i)}}{\boldsymbol{u}_j^{(i)} - \boldsymbol{l}_j^{(i)}} (h_j^{(i)} - \boldsymbol{l}_j^{(i)})$. Note that the lower bound $\boldsymbol{l}_j^{(i)}$ and upper bound $\boldsymbol{u}_j^{(i)}$ for each activation node $h_j^{(i)}$ are required in the LP construction given $\mathcal{C}_i$. They are typically computed by the existing cheap bounding methods like LiRPA variants (Wong & Kolter, 2018) with low cost. The tighter the intermediate bounds $(\boldsymbol{l}_j^{(i)}, \boldsymbol{u}_j^{(i)})$ are, the tighter $\underline{f}$ approximated by LP is.

## 2.3 LINEAR RELAXATION BASED PERTURBATION ANALYSIS (LiRPA)

**Bound propagation in LiRPA** We used Linear Relaxation based Perturbation Analysis (LiRPA)[1] as bound procedure in BaB to get linear upper and lower bounds of NN output w.r.t input $x \in \mathcal{C}$:

$$\underline{\mathbf{A}}x + \underline{\mathbf{b}} \leq f(x) \leq \overline{\mathbf{A}}x + \overline{\mathbf{b}}, \quad x \in \mathcal{C} \tag{2}$$

---

[1] We only use the backward mode LiRPA bounds (e.g., CROWN and DeepPoly) in this paper.

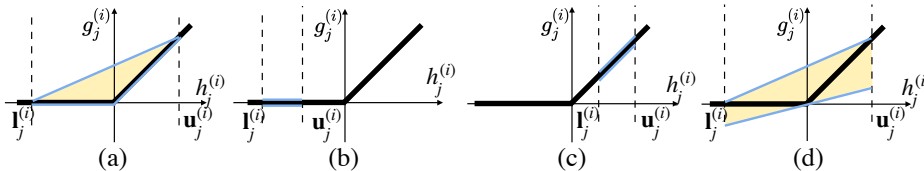

Figure 1: Relaxations of a ReLU: (a) "triangle" relaxation in LP; (b)(c) No relaxation when $\mathbf{u}_j^{(i)} \leq 0$ (always inactive) or $\mathbf{l}_j^{(i)} \geq 0$ (always active); (d) linear relaxation in LiRPA when $\mathbf{l}_j^{(i)} < 0, \mathbf{u}_j^{(i)} > 0$ (unstable).

A lower bound $\underline{f}$ can then be simply obtained by taking the lower bound of the linear equation $\underline{\mathbf{A}}x + \underline{\mathbf{b}}$ w.r.t input $x \in \mathcal{C}$, which can be obtained via Hölder's inequality when $\mathcal{C}$ is a $\ell_p$ norm ball.

To get the coefficients $\underline{\mathbf{A}}, \overline{\mathbf{A}}, \underline{\mathbf{b}}, \overline{\mathbf{b}}$, LiRPA propagates bounds of $f(x)$ as a linear function to the output of each layer, in a backward manner. At the output layer $h^{(L)}(x)$ we simply have:

$$\mathbf{I}h^{(L)}(x) \leq f(x) \leq \mathbf{I}h^{(L)}(x), \quad x \in \mathcal{C} \tag{3}$$

Then, the next step is to backward propagate the identity linear relationship through a linear layer $h^{(L)}(x) = \mathbf{W}^{(L)}g^{(L-1)}(x)$ to get the linear bounds of $f(x)$ w.r.t $g^{(L-1)}$:

$$\mathbf{W}^{(L)}g^{(L-1)}(x) \leq f(x) \leq \mathbf{W}^{(L)}g^{(L-1)}(x), \quad x \in \mathcal{C} \tag{4}$$

To get the linear relationship of $h^{(L-1)}$ w.r.t $f(x)$, we need to backward propagate through ReLU layer $g^{(L-1)}(x) = \text{ReLU}(h^{(L-1)}(x))$. Since it is nonlinear, we perform linear relaxations. For illustration, considering the $j$-th ReLU neuron at $i$-th layer, $g_j^{(i)}(x) = \text{ReLU}(h_j^{(i)}(x))$, we can linearly upper and lower bound it by $\underline{a}_j^{(i)}h_j^{(i)}(x) + \underline{b}_j^{(i)} \leq g_j^{(i)}(x) \leq \overline{a}_j^{(i)}h_j^{(i)}(x) + \overline{b}_j^{(i)}$, where $\underline{a}_j^{(i)}, \overline{a}_j^{(i)}, \underline{b}_j^{(i)}, \overline{b}_j^{(i)}$ are:

$$\begin{cases} \underline{a}_j^{(i)} = \overline{a}_j^{(i)} = 0, \underline{b}_j^{(i)} = \overline{b}_j^{(i)} = 0 & \mathbf{u}_j^{(i)} \leq 0 \quad \text{(always inactive for } x \in \mathcal{C}) \\ \underline{a}_j^{(i)} = \overline{a}_j^{(i)} = 1, \underline{b}_j^{(i)} = \overline{b}_j^{(i)} = 0 & \mathbf{l}_j^{(i)} \geq 0 \quad \text{(always active for } x \in \mathcal{C}) \\ \underline{a}_j^{(i)} = \alpha_j^{(i)}, \overline{a}_j^{(i)} = \frac{\mathbf{u}_j^{(i)}}{\mathbf{u}_j^{(i)} - \mathbf{l}_j^{(i)}}, \underline{b}_j^{(i)} = 0, \overline{b}_j^{(i)} = -\frac{\mathbf{u}_j^{(i)}\mathbf{l}_j^{(i)}}{\mathbf{u}_j^{(i)} - \mathbf{l}_j^{(i)}} & \mathbf{l}_j^{(i)} < 0, \mathbf{u}_j^{(i)} > 0 \quad \text{(unstable for } x \in \mathcal{C}) \end{cases} \tag{5}$$

Here $\mathbf{l}_j^{(i)} \leq h_j^{(i)}(x) \leq \mathbf{u}_j^{(i)}$ are intermediate pre-activation bounds for $x \in \mathcal{C}$, and $\alpha_j^{(i)}$ is an arbitrary value between 0 and 1. The pre-activation bounds $\mathbf{l}_j^{(i)}$ and $\mathbf{u}_j^{(i)}$ can be computed by treating $h_j^{(i)}(x)$ as the output neuron with LiRPA. Figure 1(b,c,d) illustrate the relaxation for each state of ReLU neuron. With these linear relaxations, we can get the linear equation of $h^{(L-1)}$ w.r.t output $f(x)$:

$$\mathbf{W}^{(L)}\underline{\mathbf{D}}_\alpha^{(L-1)}h^{(L-1)}(x) + \underline{\mathbf{b}}^{(L)} \leq f(x) \leq \mathbf{W}^{(L)}\overline{\mathbf{D}}_\alpha^{(L-1)}h^{(L-1)}(x) + \overline{\mathbf{b}}^{(L)}, \quad x \in \mathcal{C}$$

$$\underline{\mathbf{D}}_{\alpha,(j,j)}^{(L)} = \begin{cases} \underline{a}_j^{(L)}, & \mathbf{W}_j^{(L)} \geq 0 \\ \overline{a}_j^{(L)}, & \mathbf{W}_j^{(L)} < 0 \end{cases}, \quad \underline{\mathbf{b}}^{(L)} = \underline{\mathbf{b}}'^{(L)\top}\mathbf{W}^{(L)}, \quad \text{where } \underline{\mathbf{b}}_j'^{(L)} = \begin{cases} \underline{b}_j^{(L)}, & \mathbf{W}_j^{(L)} \geq 0 \\ \overline{b}_j^{(L)}, & \mathbf{W}_j^{(L)} < 0 \end{cases} \tag{6}$$

The diagonal matrices $\underline{\mathbf{D}}_\alpha^{(L-1)}, \overline{\mathbf{D}}_\alpha^{(L-1)}$ and biases reflects the linear relaxations and also considers the signs in $\mathbf{W}^{(L)}$ to maintain the lower and upper bounds. The definitions for $j$-th diagonal element $\overline{\mathbf{D}}_{\alpha,(j,j)}^{(L)}$ and bias $\overline{\mathbf{b}}^{(L)}$ are similar, with the conditions for checking the signs of $\mathbf{W}_j^{(L)}$ swapped. Importantly, $\underline{\mathbf{D}}_\alpha^{(L-1)}$ has free variables $\alpha_j^{(i)} \in [0,1]$ which do not affect correctness of the bounds. We can continue backward propagating these bounds layer by layer (e.g., $g^{(L-2)}(x), h^{(L-2)}(x)$, etc) until reaching $g^{(0)}(x) = x$, getting the eventual linear equations of $f(x)$ in terms of input $x$:

$$\mathbf{L}(x, \boldsymbol{\alpha}) \leq f(x) \leq \mathbf{U}(x, \boldsymbol{\alpha}), \quad \forall x \in \mathcal{C}, \quad \text{where}$$

$$\mathbf{L}(x, \boldsymbol{\alpha}) = \mathbf{W}^{(L)}\underline{\mathbf{D}}_\alpha^{(L-1)} \cdots \underline{\mathbf{D}}_\alpha^{(1)}\mathbf{W}^{(1)}x + \underline{\mathbf{b}}, \quad \mathbf{U}(x, \boldsymbol{\alpha}) = \mathbf{W}^{(L)}\overline{\mathbf{D}}_\alpha^{(L-1)} \cdots \overline{\mathbf{D}}_\alpha^{(1)}\mathbf{W}^{(1)}x + \overline{\mathbf{b}} \tag{7}$$

Here $\boldsymbol{\alpha}$ denotes $\alpha_j^{(i)}$ for all unstable ReLU neurons in NN. The obtained bounds $(\mathbf{L}(x, \boldsymbol{\alpha}), \mathbf{U}(x, \boldsymbol{\alpha}))$ of $f(x)$ are linear functions in terms of $x$. Beyond the simple feedforward NN presented here, LiRPA can support more complicated NN architectures like DenseNet and Transformers by computing $\mathbf{L}$ and $\mathbf{U}$ automatically and efficiently on general computational graphs (Xu et al., 2020).

**Soundness of LiRPA** The above backward bound propagation process guarantees that $\mathbf{L}(x, \alpha)$ and $\mathbf{U}(x, \alpha)$ soundly bound $f(x)$ for all $x \in \mathcal{C}$. Detailed proofs can be found in (Zhang et al., 2018; Singh et al., 2019b) for feedforward NNs and (Xu et al., 2020) for general networks.

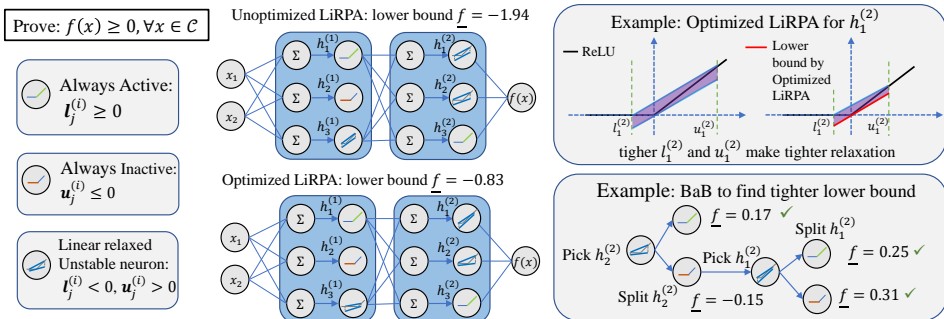

Figure 2: Illustration of our optimized LiRPA bounds and the BaB process. Given a two-layer neural network, we aim to verify output $f(x) \geq 0$. Optimized LiRPA chooses optimized slopes for ReLU lower bounds, allowing tightening the intermediate layer bounds $\boldsymbol{l}_j^{(i)}$ and $\boldsymbol{u}_j^{(i)}$ and also the output layer lower bound $\underline{f}$. BaB splits two unstable neurons $h_2^{(2)}$ and $h_1^{(2)}$ to improve $\underline{f}$ and verify all sub-domains ($\underline{f} \geq 0$ for all cases).

## 3 PROPOSED ALGORITHM

**Overview**  In this section, we will first introduce our proposed efficient optimization of LiRPA bounds on GPUs that can allow us to achieve tight approximation on par with LP or even tighter for some cases but in a much faster manner. In Fig. 2, we provide a two-layer NN example to illustrate how our optimized LiRPA can improve the performance of BaB verification. In Section 3.2, we then show that feasibility checking is important to guarantee the completeness of BaB, and BaB using LiRPA without feasibility checking will end up to be incomplete. To ensure completeness, we design our algorithm with minimal usage of LP for checking feasibility of splits. Finally, we propose a batch split design by solving a batch of sub-domains in a massively parallel manner on GPUs to fully leverage the benefits of cheap and parallelizable LiRPA. We further improve BaBSR in a parallel fashion for branching and we summarize the detailed proposed algorithm in Section 3.4.

### 3.1 OPTIMIZED LiRPA FOR COMPLETE VERIFICATION

**Concrete outer bounds with optimizable parameters**  We propose to use LiRPA as the bounding step in BaB. A pair of sound and concrete lower bound and upper bound $(\underline{f}, \overline{f})$ to $f(x)$ can be obtained according to Eq. 7 given fixed $\boldsymbol{\alpha} = \boldsymbol{\alpha}_0$:

$$\underline{f}(\boldsymbol{\alpha}_0) = \min_{x \in \mathcal{C}} \mathbf{L}(x, \boldsymbol{\alpha}_0), \quad \overline{f}(\boldsymbol{\alpha}_0) = \max_{x \in \mathcal{C}} \mathbf{U}(x, \boldsymbol{\alpha}_0) \tag{8}$$

Because $\mathbf{L}, \mathbf{U}$ are linear functions w.r.t $x$ when $\boldsymbol{\alpha}_0$ is fixed, it is easy to solve Eq. 8 using Hölder's inequality when $\mathcal{C}$ is a $\ell_p$ norm ball (Xu et al., 2020). In incomplete verification settings, $\boldsymbol{\alpha}$ can be set via certain heuristics (Zhang et al., 2018). Salman et al. (2019) showed that, the variable $\boldsymbol{\alpha}$ is equivalent to dual variables in the LP relaxed verification problem (Wong & Kolter, 2018). Thus, an optimal selection of $\boldsymbol{\alpha}$ given the *same* pre-activation bounds $\mathbf{l}_j^{(i)}$ and $\mathbf{u}_j^{(i)}$ can in fact, lead to the the same optimal solution for $\underline{f}$ and $\overline{f}$ as in LP.

Previous complete verifiers typically use LiRPA variants to obtain intermediate layer bounds to construct an LP problem (Bunel et al. (2018); Lu & Kumar (2020)) and solve the LP to obtain bounds at output layer. The main reason for using LP is that it typically produces much tighter bounds than LiRPA when $\boldsymbol{\alpha}$ is not optimized. We use optimized LiRPA, which is fast, accelerator-friendly, and can produce tighter bounds, well outperforming LP for complete verification:

$$\underline{f} = \min_{\boldsymbol{\alpha}} \min_{x \in \mathcal{C}} \mathbf{L}(x, \boldsymbol{\alpha}), \quad \overline{f} = \max_{\boldsymbol{\alpha}} \max_{x \in \mathcal{C}} \mathbf{U}(x, \boldsymbol{\alpha}), \quad \alpha_j^{(i)} \in [0, 1] \tag{9}$$

The inner minimization or maximization has closed form solutions (Xu et al., 2020) based on Hölder's inequality, so we only need to optimize on $\boldsymbol{\alpha}$. Since we use a differentiable framework (Xu et al., 2020) to compute the LiRPA bound functions $\mathbf{L}$ and $\mathbf{U}$, the gradients $\frac{\partial \mathbf{L}}{\partial \boldsymbol{\alpha}}$ and $\frac{\partial \mathbf{U}}{\partial \boldsymbol{\alpha}}$ can be obtained easily. Optimization over $\boldsymbol{\alpha}$ can be done via projected gradient descent (each coordinate of $\boldsymbol{\alpha}$ is constrained in $[0, 1]$). Since the gradient computation and optimization are done on GPUs, the bounding process is still very fast and can be one or two magnitudes faster than solving an LP.

**Optimized LiRPA bounds can be tighter than LP**  Solving Eq. 9 using gradient descent cannot guarantee to converge to the global optima, so it seems the bounds must be looser than LP. Counter-intuitively, by optimizing $\boldsymbol{\alpha}$, we can potentially *obtain tighter bounds* than LP. When a "triangle"

relaxation is constructed for LP, intermediate pre-activation bounds $\mathbf{l}_j^{(i)}$, $\mathbf{u}_j^{(i)}$ must be fixed for the $j$-th ReLU in layer $i$. During the LP optimization process, only the output bounds are optimized; intermediate bounds stay unchanged. However, in the LiRPA formulation, $\mathbf{L}(x, \boldsymbol{\alpha})$ and $\mathbf{U}(x, \boldsymbol{\alpha})$ are complex functions of $\boldsymbol{\alpha}$: since intermediate bounds are also computed by LiRPA, they depend on all $\alpha_{j'}^{(i')}(0 < i' < i)$ in previous layers. Thus, the gradients $\frac{\partial \mathbf{L}}{\partial \boldsymbol{\alpha}}$ and $\frac{\partial \mathbf{U}}{\partial \boldsymbol{\alpha}}$ can tighten output layer bounds $\underline{f}$ and $\overline{f}$ indirectly by tightening intermediate layer bounds, forming a tighter convex relaxation for the next iteration. An LP solver cannot achieve this because adding $\mathbf{l}_j^{(i)}$ and $\mathbf{u}_j^{(i)}$ as optimization variables makes the problem non-linear. This is the key to our success of applying LiRPA based bounds for the complete verification setting, where tighter bounds are essential.

In Figure 3, we illustrate our optimized LiRPA bounds and the LP solution. Initially, we use LiRPA with $\boldsymbol{\alpha}$ set via a fast heuristic to compute intermediate layer bounds $\mathbf{l}_j^{(i)}$ and $\mathbf{u}_j^{(i)}$, and then use them to build a relaxed LP problem. The solution to this initial LP problem (red line) is much tighter than the LiRPA solution with the heuristically set $\boldsymbol{\alpha}$ (the left-most point of the blue line). Then, we optimize $\boldsymbol{\alpha}$ with gradient decent, and LiRPA quickly outperforms this initial LP solution due to optimized tighter intermediate layer bounds. We can create a new LP with optimized intermediate bounds (light blue line), producing a slightly tighter bound than LiRPA with optimized $\boldsymbol{\alpha}$. The LP bounds in most existing complete verifiers all use

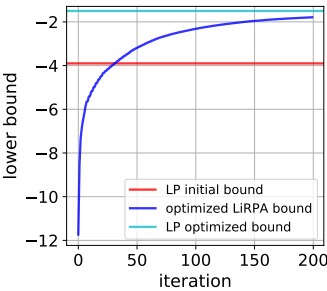

Figure 3: Optimized LiRPA bound (0 to 200 iterations) vs LP bounds.

intermediate layer bounds obtained from unoptimized LiRPA bounds or even weaker methods like interval arithmetic, ending up to the solution close to or lower than the red line in Figure 3. Instead, our optimized LiRPA bounds can produce tight bounds, and also exploit parallel acceleration from machine learning accelerators, leading to huge improvements in verification time compared to existing baselines.

**ReLU Split Constraints** In the BaB process, when a ReLU $h_j^{(i)}$ is split into two sub-domains ($h_j^{(i)} \geq 0$ and $h_j^{(i)} < 0$), we simply set $\boldsymbol{l}_j^{(i)} \geq 0$ and $\boldsymbol{u}_j^{(i)} < 0$ in bounding step. It tighten the LiRPA bounds by forcing the split ReLU linear, reducing relaxation errors. However, when splits are added, LiRPA and LP are not equivalent even under fixed $\mathbf{l}_j^{(i)}$, $\mathbf{u}_j^{(i)}$ and optimal $\boldsymbol{\alpha}$. After splits, LiRPA cannot check certain constraints where LP is capable to, as we will discuss in the next section.

## 3.2 COMPLETENESS WITH MINIMAL USAGE OF LP BOUNDING PROCEDURE

Even though our optimized LiRPA can bring us huge speed improvement over LP for BaB based verification, we observe that it may end up to be incomplete due to the lack of feasibility checking: it cannot detect some conflicting settings of ReLU splits. We state such an observation in Theorem 3.1:

**Theorem 3.1 (Incompleteness without feasibility checking)** *When using LiRPA variants described in Section 2.3 as the bounding procedure, BaB based verification is incomplete.*

We prove the theorem by giving a counter-example in Appendix A.1 where *all* ReLU neurons are split and thus LiRPA runs on a linear network for each sub-domain. As a result, LiRPA can still be indecisive for the verification problem. The main reason is that LiRPA variants will lose the feasibility information encoded by the sub-domain constraints. For illustration, consider a sub-domain $\mathcal{C}_i = C \cap (h_{j_1}^{(i_1)} < 0) \cap (h_{j_2}^{(i_2)} \geq 0)$, LiRPA will force $g_{j_1}^{(i_1)}(x) = 0$ (inactive ReLU, a zero function) and $g_{j_2}^{(i_2)}(x) = h_{j_2}^{(i_2)}(x)$ (active ReLU, an identity function) respectively and propagate these bounds to get the approximated lower bound $\underline{f}_{\mathcal{C}_i}$. However, the split feasibility constraint $(h_{j_1}^{(i_1)} < 0) \cap (h_{j_2}^{(i_2)} \geq 0)$ is ignored, so two conflict splits may be conducted (e.g., when $h_{j_1}^{(i_1)} < 0$, $h_{j_2}^{(i_2)}$ cannot be $\geq 0$). On the contrary, LP can fully preserve such feasibility information due to the linear solver involved and detect the infeasible sub-domains. Then, in Theorem 3.2 we show that the minimal usage of feasibility checking with LP can guarantee the completeness of BaB with LiRPA.

---

**Algorithm 1** Parallel BaB with optimized LiRPA bounding (we highlight the differences between our algorithm and regular BaB (Bunel et al., 2018) in blue. Comments are in brown.)

---

1: **Inputs**: $f$, $\mathcal{C}$, $n$ (batch size), $\eta$ (threshold to switch to LP)
2: $(\underline{f}, \overline{f}) \leftarrow \texttt{optimized\_LiRPA}(f, [\mathcal{C}])$
3: $\mathbb{P} \leftarrow \left[(\underline{f}, \overline{f}, \mathcal{C})\right]$ ▷ $\mathbb{P}$ is the set of all unverified sub-domains
4: **while** $\underline{f} < 0$ **and** $\overline{f} \geq 0$ **do**
5:     $(\mathcal{C}_1, \ldots, \mathcal{C}_n) \leftarrow \texttt{batch\_pick\_out}(\mathbb{P}, n)$ ▷ Pick sub-domains to split and removed them from $\mathbb{P}$
6:     $\left[\mathcal{C}_1^l, \mathcal{C}_1^u, \ldots, \mathcal{C}_n^l, \mathcal{C}_n^u\right] \leftarrow \texttt{batch\_split}(\mathcal{C}_1, \ldots, \mathcal{C}_n)$ ▷ Each $\mathcal{C}_i$ splits into two sub-domains $\mathcal{C}_i^l$ and $\mathcal{C}_i^u$
7:     $\left[\underline{f}_{\mathcal{C}_1^l}, \overline{f}_{\mathcal{C}_1^l}, \underline{f}_{\mathcal{C}_1^u}, \overline{f}_{\mathcal{C}_1^u}, \ldots, \underline{f}_{\mathcal{C}_n^l}, \overline{f}_{\mathcal{C}_n^l}, \underline{f}_{\mathcal{C}_n^u}, \overline{f}_{\mathcal{C}_n^u}\right] \leftarrow \texttt{optimized\_LiRPA}(f, [\mathcal{C}_1^l, \mathcal{C}_1^u, \ldots, \mathcal{C}_n^l, \mathcal{C}_n^u])$ ▷
    Compute lower and upper bounds using LiRPA for each sub-domain on GPUs in a batch
8:     $\mathbb{P} \leftarrow \mathbb{P} \bigcup \texttt{Domain\_Filter}\left([\underline{f}_{\mathcal{C}_1^l}, \overline{f}_{\mathcal{C}_1^l}, \mathcal{C}_1^l], [\underline{f}_{\mathcal{C}_1^u}, \overline{f}_{\mathcal{C}_1^u}, \mathcal{C}_1^u], \ldots, [\underline{f}_{\mathcal{C}_n^l}, \overline{f}_{\mathcal{C}_n^l}, \mathcal{C}_n^l], [\underline{f}_{\mathcal{C}_n^u}, \overline{f}_{\mathcal{C}_n^u}, \mathcal{C}_n^u]\right)$ ▷
    Filter out verified sub-domains, insert the left domains back to $\mathbb{P}$
9:     $\underline{f} \leftarrow \min\{\underline{f}_{\mathcal{C}_i} \mid (\underline{f}_{\mathcal{C}_i}, \mathcal{C}_i) \in \mathbb{P}\}, i = 1, \ldots, n$ ▷ To ease notation, $\mathcal{C}_i$ here indicates both $\mathcal{C}_i^u$ and $\mathcal{C}_i^l$
10:     $\overline{f} \leftarrow \min\{\overline{f}_{\mathcal{C}_i} \mid (\overline{f}_{\mathcal{C}_i}, \mathcal{C}_i) \in \mathbb{P}\}, i = 1, \ldots, n$
11:     **if** $\texttt{length}(\mathbb{P}) > \eta$ **then** ▷ Fall back to LP for completeness
12:         $\left[\underline{f}_{\mathcal{C}_1^l}, \overline{f}_{\mathcal{C}_1^l}, \underline{f}_{\mathcal{C}_1^u}, \overline{f}_{\mathcal{C}_1^u}, \ldots, \underline{f}_{\mathcal{C}_n^l}, \overline{f}_{\mathcal{C}_n^l}, \underline{f}_{\mathcal{C}_n^u}, \overline{f}_{\mathcal{C}_n^u}, \right] \leftarrow \texttt{compute\_bound\_LP}(f, [\mathcal{C}_1^l, \mathcal{C}_1^u, \ldots, \mathcal{C}_n^l, \mathcal{C}_n^u])$
13:         $\mathbb{P} \leftarrow \mathbb{P} \bigcup \texttt{Domain\_Filter}\left([\underline{f}_{\mathcal{C}_1^l}, \overline{f}_{\mathcal{C}_1^l}, \mathcal{C}_1^l], [\underline{f}_{\mathcal{C}_1^u}, \overline{f}_{\mathcal{C}_1^u}, \mathcal{C}_1^u], \ldots, [\underline{f}_{\mathcal{C}_n^l}, \overline{f}_{\mathcal{C}_n^1}, \mathcal{C}_n^l], [\underline{f}_{\mathcal{C}_n^u}, \overline{f}_{\mathcal{C}_n^u}, \mathcal{C}_n^u]\right)$
14: **Outputs:** $\underline{f}, \overline{f}$

---

**Theorem 3.2 (Minimal feasibility checking for completeness)** *When using LiRPA variants described in Section 2.3 as the bounding procedure, BaB based verification is complete if all infeasible leaf sub-domains (i.e., sub-domains cannot be further split) are detected by linear programming.*

We prove the theorem in Appendix A.2, where we show that by checking the feasibility of splits with LP, we can eliminate the cases where incompatible splits are chosen in the LiRPA BaB process. Since LP is slow while LiRPA is highly efficient, we propose to only use LP when the LiRPA based bounding process is stuck, either (1) when partitioning and bounding new sub-domains with LiRPA cannot further improve the bounds, or (2) when all unstable neurons have been split. In this way, the infeasible sub-domains can be eventually detected by occasional usage of LP while the advantage of massive parallel LiRPA on GPUs is fully enjoyed. We will describe our full algorithm in Sec. 3.4.

### 3.3 BATCH SPLITS

SOTA BaB methods (Bunel et al., 2020b; Lu & Kumar, 2020) only split one sub-domain during each branching step. Since we use cheap and GPU-friendly LiRPA bounds, we can select a batch of sub-domains to split and propagate their LiRPA bounds in a batch. Such a batch splitting design can greatly improve hardware efficiency on GPUs. Given a batch size $n$ that allows us to fully use the GPU memory available, we can obtain $n$ bounds simultaneously. It grows the search tree on a single leaf by a depth of $\log_2 n$, or split $n/2$ leaf nodes at the same time, accelerating by up to $n$ times.

### 3.4 OUR COMPLETE VERIFICATION FRAMEWORK

Our LiRPA based complete verification framework is presented in Alg. 1. The algorithm takes a target NN function $f$ and a domain $\mathcal{C}$ as inputs. We run optimized LiRPA to get initial bounds $(\underline{f}, \overline{f})$ for $x \in \mathcal{C}$ (Line 2). Then we utilize the power of GPUs to split in parallel and maintain a global set $\mathbb{P}$ storing all the sub-domains which cannot be verified with optimized LiRPA (Line 5-10). Specifically, $\texttt{batch\_pick\_out}$ improves BaBSR (Bunel et al., 2018) in a parallel manner to select $n$ sub-domains in $\mathbb{P}$ and determine the corresponding ReLU neuron to split for each of them. If the length of $\mathbb{P}$ is less than $n$, then we reduce $n$ to the length of $\mathbb{P}$. $\texttt{batch\_split}$ splits each selected $\mathcal{C}_i$ to two sub-domains $\mathcal{C}_i^l$ and $\mathcal{C}_i^u$ by forcing the selected unstable ReLU neuron to be positive and negative, respectively. $\texttt{optimize\_LiRPA}$ runs optimized LiRPA in parallel as a batch and returns the lower and upper bounds for $n$ selected sub-domains simultaneously. $\texttt{Domain\_Filter}$ filters out verified sub-domains (proved with $\underline{f}_{\mathcal{C}_i} \geq 0$) and we insert the remaining ones to $\mathbb{P}$. The loop breaks if the property is proved ($\underline{f} \geq 0$) or a counter-example is found in any sub-domain ($\overline{f} < 0$).

To avoid excessive splits, we set the maximum length of the sub-domains to $\eta$ (Line 12). Once the length of $\mathbb{P}$ reaches this threshold, $\texttt{compute\_bound\_LP}$ will be called. It solves these $\eta$ sub-domains by LP (one by one in a loop, or in parallel if using multiple CPUs is allowed) with optimized LiRPA

computed intermediate layer bounds. If a sub-domain $\mathcal{C}_i \in \mathbb{P}$ (which previously cannot be verified by LiRPA) is proved or detected to be infeasible by LP, as an effective heuristic, we will backtrack and prioritize to check its parent node with LP. If the parent sub-domain is also proved or infeasible, we can prune all its child nodes to greatly reduce the size of the search tree.

**Completeness of our framework** Our algorithm is *complete*, because we follow Theorem 3.2 and check feasibility of all split sub-domains that have deep BaB search tree depth (length of $\mathbb{P}$ reaches threshold $\eta$), forming a superset of the worst case where all ReLU neurons are split.

## 4 EXPERIMENTS

In this section, we compare our verifier against the state-of-the-art ones to illustrate the effectiveness of our proposed framework. Overall, our verifier is about 10X, 4X and 20X faster than the best LP-based verifier (Lu & Kumar, 2020) on the Base, Wide and Deep models, respectively.

**Setup** We follow the most challenging experimental setup used in the state-of-the-art verifiers GNN-ONLINE (Lu & Kumar, 2020) and BABSR (Bunel et al., 2020b). Specifically, we evaluate on CIFAR10 dataset on three NNs: Base, Wide and Deep. The dataset is categorized into three difficulty levels: Easy, Medium, and Hard, which is generated according to the performance of BaBSR. The verification task is defined as given a $l_\infty$ norm perturbation less than $\epsilon$, the classifier will not predict a specific (predefined) wrong label for each image $x$ (see Appendix B). We set batch size $n = 400, 200, 200$ for Base, Wide and Deep model respectively and threshold $\eta = 12000$. More details on experimental setup are provided in Appendix B. Our code is available at https://github.com/kaidixu/LiRPA_Verify.

**Comparisons against state-of-the-art verifiers** We include five different methods for comparison: (1) BABSR (Bunel et al., 2020b), a BaB and LP based verifier using a simple ReLU split heuristic; (2) MIPPLANET (Ehlers, 2017), a customized MIP solver for NN verification where unstable ReLU neurons are randomly selected for split; (3) GNN (Lu & Kumar, 2020) and (4) GNN-ONLINE (Lu & Kumar, 2020) are BaB and LP based verifiers using a learned graph neural network (GNN) to guide the ReLU split. (5) PROXIMAL BABSR (Bunel et al., 2020a) is a very recently proposed verification framework based on Lagrangian decomposition which also supports GPU acceleration without solving LPs. All methods use 1 CPU with 1 GPU. The timeout threshold is 3,600 seconds.

For the Base model in different difficulty levels, Easy, Medium and Hard, Table 1 shows that we are around $5 \sim 40$X faster than baseline BaBSR and around $2 \sim 20$X faster than GNN split baselines. The accumulative solved properties with increasing runtime are shown in Figure 4. In all our experiments, we use the basic heuristic in BaBSR for branching and do not use GNNs, so our speedup comes purely from the faster LiRPA based bounding procedure. We are also competitive against Lagrangian decomposition on GPUs.

Table 1: Performance of various methods on different models. We compare each method's avg. solving time, the avg. number of branches required, and the percentage of timed out (TO) properties.

| Method | Base - Easy | | | Base - Medium | | | Base - Hard | | | Wide | | | Deep | | |
|---|---|---|---|---|---|---|---|---|---|---|---|---|---|---|---|
| | time(s) | branches | %TO | time(s) | branches | %TO | time(s) | branches | %TO | time(s) | branches | %TO | time(s) | branches | %TO |
| BABSR | 522.48 | 585 | 0.0 | 1335.40 | 1471 | 0.0 | 2875.16 | 1843 | 35.2 | 3325.65 | 455 | 50.3 | 2855.19 | 365 | 54.0 |
| MIPPLANET | 1462.24 | - | 16.5 | 1912.25 | - | 43.5 | 2172.23 | - | 46.2 | 3088.40 | - | 79.4 | 2842.54 | - | 73.6 |
| GNN | 312.93 | 301 | 0.0 | 624.12 | 635 | 0.9 | 1468.75 | 931 | 15.6 | 1791.52 | 375 | 19.0 | 1870.63 | 198 | 18.4 |
| GNN-ONLINE | 207.43 | 269 | 0.0 | 638.15 | 546 | 0.4 | 1255.35 | 968 | 15.6 | 1642.03 | 389 | 19.0 | 1845.71 | 196 | 18.4 |
| PROXIMAL BABSR | 15.68 | 1371 | 0.0 | 51.88 | 6482 | 0.4 | **627.96** | 91880 | 13.4 | 510.55 | 45855 | 11.4 | 230.06 | 6721 | 4.4 |
| OURS | **11.86** | 2589 | 0.0 | **42.04** | 9233 | **0.0** | 633.85 | 96755 | **13.0** | **375.23** | 53481 | **8.5** | **81.55** | 1439 | **1.6** |

Figure 4: Cactus plots for our method and other baselines in Base (Easy, Medium and Hard ), Wide and Deep models. We plot the percentage of solved properties with growing running time.

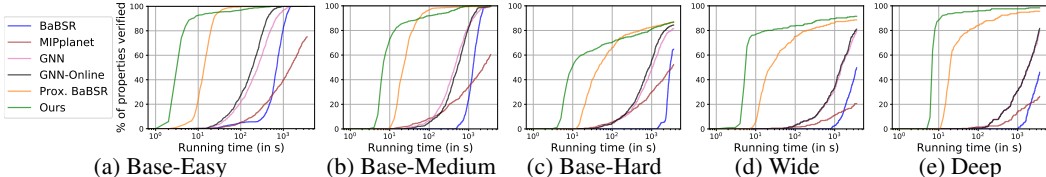

(a) Base-Easy    (b) Base-Medium    (c) Base-Hard    (d) Wide    (e) Deep

**Performance on Larger Models** In Table 1, we show that our verifier is more scalable on larger (wider or deeper) NNs compared to other state-of-the-art verifiers. Our method enjoys efficient GPU acceleration particularly on Deep model and can achieve 30X speedup compared to BABSR,

and we are also significantly faster than Lagrangian decomposition based GPU verifier (PROXIMAL BABSR). When compared to the state-of-the-art LP based BaB, GNN-ONLINE, our method can save 20X running time on Deep model. In Appendix C, we analyze the effectiveness of optimized LiRPA and batch splits separately, and find that optimized LiRPA is crucial for NN verification. Performance comparisons of our proposed framework on CPU cores without GPU acceleration are included in Appendix D.

## 5 RELATED WORK

**Complete verifiers** Early complete verifiers rely on satisfiability modulo theory (SMT) (Katz et al., 2017; Huang et al., 2017; Ehlers, 2017) and mixed integer linear programming (MILP) (Tjeng et al., 2019a; Dutta et al., 2018), and they typically do not scale well. Higher order logic provers such as proof assistant (Bentkamp et al., 2018) can also be potentially used for NN verification, but their scalability to the NN setting has not been demonstrated. Recently, Bunel et al. (2018) unified many approaches used in various complete verifiers into a BaB framework. An LP based bounding procedure is used in most of the existing BaB framework (Bunel et al., 2018; Wang et al., 2018c; Royo et al., 2019; Lu & Kumar, 2020). For branching, two categories of branching strategies were proposed: (1) input node branching (Wang et al., 2018c; Bunel et al., 2020b; Royo et al., 2019; Anderson et al., 2019) where input features are divided into sub-domains, and (2) activation node (especially, ReLU) branching (Katz et al., 2017; Bunel et al., 2018; Wang et al., 2018b; Ehlers, 2017; Lu & Kumar, 2020) where hidden layer activations are split into sub-domains. Bunel et al. (2018) found that input node branching cost is exponential to input dimension. Thus, many state-of-the-art verifiers use activation node branching instead, focusing on heuristics to select good nodes to split. BaBSR (Bunel et al., 2018) prioritizes ReLUs for splitting based on their pre-activation bounds; Lu & Kumar (2020) used a graph neural network (GNN) to learn good splitting heuristics. Our work focuses on improving bounding and can use better branching heuristics to achieve further speedup.

Two mostly relevant concurrent works using GPUs for accelerating NN verification are: (1) GPUPoly (Müller et al., 2020), an extension of DeepPoly on CUDA, is *still an incomplete verifier*. Also, it is implemented in CUDA C++, requiring manual effort for customization and gradient computation, so it is not easy to get the gradients for optimizing bounds as we have done in Section 3.1. (2) Lagrangian Decomposition (Bunel et al., 2020a) is a GPU-accelerated BaB based complete verifier that iteratively tightens the bounds based on a Lagrangian decomposition optimization formulation and does not reply on LP. However, it solves a much more complicated optimization problem than LiRPA, and typically requires hundreds of iterations to converge for a single sub-domain.

**Incomplete verifiers** Many incomplete verification methods rely on convex relaxations of NN, replacing nonlinear activations like ReLUs with linear constraints (Wong & Kolter, 2018; Wang et al., 2018b; Zhang et al., 2018; Weng et al., 2018; Gehr et al., 2018; Singh et al., 2018a;b; 2019b;a) or semidefinite constraints (Raghunathan et al., 2018; Dvijotham et al., 2020; Dathathri et al., 2020). Tightening the relaxation for incomplete verification was discussed in (Dvijotham et al., 2018; Singh et al., 2019a; Lyu et al., 2019; Tjandraatmadja et al., 2020). Typically, tight relaxations require more computation and memory in general. We refer the readers to (Salman et al., 2019) for a comprehensive survey. Recently, Xu et al. (2020) categorized the family of linear relaxation based incomplete verifiers into LiRPA framework, allowing efficient implementation on machine learning accelerators. Our work uses LiRPA as the bounding procedure for complete verification and exploits its computational efficiency to accelerate, and our main contribution is to show that we can use fast but weak incomplete verifiers as the main driver for complete verification when strategically applied.

## 6 CONCLUSION

We use a LiRPA based incomplete NN verifier to accelerate the bounding procedure in branch and bound (BaB) for complete NN verification on massively parallel accelerators. We use a fast gradient based procedure to tighten LiRPA bounds. We study the completeness of BaB with LiRPA, and show up to 5X speedup compared to state-of-the-art verifiers across multiple models and properties.

## ACKNOWLEDGMENTS

This work is supported by NSF grant CNS18-01426; an ARL Young Investigator (YIP) award; an NSF CAREER award; a Google Faculty Fellowship; a Capital One Research Grant; and a J.P. Mor-

gan Faculty Award; Air Force Research Laboratory under FA8750-18-2-0058; NSF IIS-1901527 and NSF IIS-2008173.

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

# A  PROOFS

## A.1  PROOF OF THEOREM 3.1

We prove Theorem 3.1 by providing a simple counterexample, and we illustrate the necessity of feasibility checking for the completeness of BaB based verification. Consider an NN with only two ReLU units $g_1^{(2)} = \text{ReLU}(h_1^{(2)})$ and $g_2^{(2)} = \text{ReLU}(h_2^{(2)})$ where they share the same one dimension input $h_1^{(2)} = h_2^{(2)} = x$. The final output function of NN is defined as $f = g_1^{(2)} - g_2^{(2)}$. As a verification problem, we want to verify the property $f \geq 0$ where $x = [-1, 1]$. Since hidden nodes $h_1$ and $h_2$ are exactly the same, the ground-truth output range is $f^*(x) \in [0, 0]$. A complete BaB based verifier is expected to obtain that optimal bound and prove the property after splitting $h_1$ and $h_2$ together while BaB with only LiPRA cannot guarantee that completeness. Specifically, BaB with only LiRPA will split the original domain $x \in [-1, 1]$ into four sub-domains and approximate the bound with LiRPA respectively:

(1) (feasible) sub-domain $x \in [-1, 1], h_1^{(2)} \geq 0, h_2^{(2)} \geq 0$ with output $f = [x, x] - [x, x] \in [0, 0]$

(2) (feasible) sub-domain $x \in [-1, 1], h_1^{(2)} < 0, h_2^{(2)} < 0$ with output $f = [0, 0] - [0, 0] \in [0, 0]$

(3) (infeasible) sub-domain $x \in [-1, 1], h_1^{(2)} < 0, h_2^{(2)} \geq 0$ with output $f = [0, 0] - [x, x] \in [-1, 1]$

(4) (infeasible) sub-domain $x \in [-1, 1], h_1^{(2)} \geq 0, h_2^{(2)} < 0$ with output $f = [x, x] - [0, 0] \in [-1, 1]$

Only the first two split sub-domains are feasible and therefore the ground-truth lower bound $0$ can be obtained by taking the minimum of the estimated bounds from sub-domains (1) and (2). However, pure LiRPA is not able to tell the infeasibility of sub-domains (3) and (4) and thus BaB with pure LiRPA will report the minimum $-1$ got from all these four sub-domains as the global lower bound for the original input domain, ending up not being able to verify the property, i.e., incomplete.

## A.2  PROOF OF THEOREM 3.2

We prove Theorem 3.2 by considering the worst case where all unstable ReLU neurons are split.

Given a neural network function $f$ with input domain $\mathcal{C}$, assume there are $N$ unstable ReLU neurons $\{g_i = \text{ReLU}(h_i) | i = 1, \cdots, N\}$ in total. In the worst case, we have $2^N$ leaf sub-domains $\mathcal{S} = \{\mathcal{C}_i | i = 1, \cdots, 2^N\}$, where each $\mathcal{C}_i$ corresponds to one assignment of unstable ReLU neuron splits. For example, we can have

$$\mathcal{C}_1 = \mathcal{C} \cap (h_1 \geq 0) \cap (h_2 \geq 0) \cap \cdots \cap (h_N \geq 0)$$
$$\mathcal{C}_2 = \mathcal{C} \cap (h_1 < 0) \cap (h_2 \geq 0) \cap \cdots \cap (h_N \geq 0)$$
$$\mathcal{C}_3 = \mathcal{C} \cap (h_1 \geq 0) \cap (h_2 < 0) \cap \cdots \cap (h_N \geq 0)$$
$$\mathcal{C}_4 = \mathcal{C} \cap (h_1 < 0) \cap (h_2 < 0) \cap \cdots \cap (h_N \geq 0)$$
$$\cdots$$

Note that by definition the original input domain $\mathcal{C} = \cup_{\mathcal{C}' \in \mathcal{S}} \mathcal{C}'$; in other words, all the $2^N$ split sub-domains combined will be the same as the original input domain.

Not all of the sub-domains are actually feasible, due to the consistency requirements between neurons. For example, in our proof in Section A.1, $h_1^{(2)}$ and $h_2^{(2)}$ cannot be both $\geq 0$ or both $< 0$. We can divide the sub-domains $\mathcal{S}$ into two mutually exclusive sub-sets, $\mathcal{S}^{\text{feas}}$ for all the feasible sub-domains, and $\mathcal{S}^{\text{infeas}}$ for all the infeasible sub-domains. We have $\mathcal{C} = \cup_{\mathcal{C}' \in \mathcal{S}^{\text{feas}}} \mathcal{C}'$ since these infeasible sub-domains are empty sets.

We first show that linear programming (LP) can be used to effectively detect these infeasible sub-domains. For some $\mathcal{C}' \in \mathcal{S}^{\text{infeas}}$, because all the ReLU neurons are fixed to be positive or negative, no relaxation is needed and the network is essentially linear; thus, the input value of every hidden neuron $h_i$ can be written as a linear equation w.r.t. input $x$. We add all the Boolean predicates on $h_i$ to a LP problem as linear constraints w.r.t $x$. If this LP is feasible, then we can find some input $x_0$ that assigns compatible values to all $h_i$; otherwise, the LP is infeasible.

Due to the lack of feasibility checking in LiRPA, the computed global lower (or upper) bounds from LiRPA is $\underline{f}_{\text{LiRPA}} = \min_{\mathcal{C}' \in \mathcal{S}} \underline{f}_{\mathcal{C}'} = \min \left( \min_{\mathcal{C}' \in \mathcal{S}^{\text{feas}}} \underline{f}_{\mathcal{C}'}, \min_{\mathcal{C}' \in \mathcal{S}^{\text{infeas}}} \underline{f}_{\mathcal{C}'} \right)$. With feasibility

checking from LP, we can remove all infeasible sub-domains from this $\min$ such that they do not contribute to the global lower bound: $\underline{f} = \min_{\mathcal{C}' \in \mathcal{S}^{\text{feas}}} \underline{f}_{\mathcal{C}'}$.

To prove the whole BaB verification is complete, it is sufficient to prove this lower bound $\underline{f}$ is the exact minimum of $f$ bounded in $\mathcal{C}$. Since any sub-domain $\mathcal{C}' \in \mathcal{S}^{\text{feas}}$ is a leaf sub-domain with no unstable ReLU neurons, the neural network bounded within $\mathcal{C}'$ is a linear function. LiRPA can give an exact minimum of $f$ within sub-domain $\mathcal{C}'$ . Since $\mathcal{C} = \cup_{\mathcal{C}' \in \mathcal{S}^{\text{feas}}} \mathcal{C}'$ (in other words, $\mathcal{S}^{\text{feas}}$ covers all the feasible sub-domains within $\mathcal{C}$), the minimal value for all of them $\underline{f} = \min_{\mathcal{C}' \in \mathcal{S}^{\text{feas}}} \underline{f}_{\mathcal{C}'}$ forms the exact minimum of $f$ within the input domain $\mathcal{C}$. Thus, BaB with LiRPA based bounding procedure is complete when feasibility checking is applied.

## B  EXPERIMENTAL SETUP

We use the same set of models and benchmark examples used in the state-of-the-art verifiers GNN-ONLINE (Lu & Kumar, 2020) and BABSR (Bunel et al., 2020b). Specifically, we evaluate on the most challenging CIFAR-10 dataset with the same standard robustly trained convolutional neural networks: Base, Wide, and Deep. These model structures are also used in (Lu & Kumar, 2020; Bunel et al., 2020a). The Base model contains 2 convolution layers with 8 and 16 filters as well as two linear layers with 100 and 10 hidden units, respectively. In total, the Base model has 3,172 ReLU activation units. The Wide model contains 2 convolution layers with 16 and 32 filters and two linear layers with 100 and 10 hidden units, respectively, which contains 6,244 ReLU activation units in total. The Deep model contains 4 convolution layers and all of them have 8 filters and two linear layers with 100 and 10 hidden units, respectively, with 3,756 ReLU activation units in total. The source code of BABSR, MIPPLANET, GNN and GNN-ONLINE are available at `https://github.com/oval-group/GNN_branching`. The source code of PROXIMAL BABSR is available at `https://github.com/verivital/vnn-comp` by replacing the dataset to the same one we used here.

Given an correctly classified image $x$ with label $y_c$, and another wrong label $y_{c'} \neq y_c$ (pre-defined in this benchmark) and $\epsilon$, the verifier needs to prove:

$$(e^{(c)} - e^{(c')})^T f(x') > 0 \qquad \text{s.t } \forall x' \quad \|x - x'\|_\infty \leq \epsilon \tag{10}$$

where $f(\cdot)$ is the logit-layer output of a multi-class classifier, $e^{(c)}$ and $e^{(c')}$ are one-hot encoding vectors for labels $y_c$ and $y_{c'}$. We want to verify that for a given $\epsilon$, the trained classifier will not predict wrong label $y_{c'}$ for image $x$. All properties including $x$, $\epsilon$, and $c'$ are provided by (Lu & Kumar, 2020). Specifically, they categorize verification properties solved by BABSR within 800s as easy, between 800s and 2400s as medium and more than 2400s as hard.

Our experiments are conducted on one Intel I7-7700K CPU and one Nvidia GTX 1080 Ti GPU. The parallel batch size $n$ is set to 400, 200 and 200 for base, wide and deep model respectively and the $\eta$ is set to 12,000 due to GPU memory constraint. To make a fair comparison, we use one CPU core for all methods. Also, we use one GPU for GNN, GNN-ONLINE, PROXIMAL-BABSR and our method. When optimizing the LiRPA bounds, we apply 100 steps gradient decent for obtaining the initial $\underline{f}$ (Line 2 in Algorithm 1). After that, we use 10 steps gradient decent (Line 7) and early stop once $\underline{f} > 0$ or $\underline{f}$ has no improvement.

## C  ABLATION STUDY

Our efficient framework leverages two powerful components: (1) optimized LiRPA bounds and (2) batch splits on GPUs. In this section, we conduct breakdown experiments to show how each individual technique can help with complete verification. As we can see in Table 2, using batched split with unoptimized LiRPA is not very successful and cannot beat BABSR. We observe that, without optimized LiRPA, the bounds are very loose and cannot quickly improve the global lower bound. In contrast, using optimized LiRPA bounds without batch splits (splitting a single node at a time and running a batch size of 1 on GPU) can still significantly speed up complete verification, around $2 \sim 10\text{X}$ compared to BABSR. Finally, combining batch splits and optimized LiRPA allows us to gain up to 44X speedup compared to BABSR.

Table 2: Ablation study for different components of our algorithm. The speedup rate is computed based on running time of BABSR baseline: speedup = Time of BaBSR/Time of our method.

| Method | Easy time(s) | Easy speedup | Medium time(s) | Medium speedup | Hard time(s) | Hard speedup | Wide time(s) | Wide speedup | Deep time(s) | Deep speedup |
|---|---|---|---|---|---|---|---|---|---|---|
| BABSR baseline | 522.48 | – | 1335.40 | – | 2875.16 | – | 3325.65 | – | 2855.19 | – |
| Batch Splits (unoptimized LiRPA) | 587.10 | 0.89 | 1470.02 | 0.91 | 3013.57 | 0.95 | 3457.30 | 0.96 | 2998.50 | 0.95 |
| Optimized LiRPA (no batch splits) | 94.08 | 5.58 | 361.53 | 3.70 | 1384.22 | 2.07 | 736.56 | 4.51 | 287.33 | 9.94 |
| Optimized LiRPA & Batch Splits | **11.86** | 44.05 | **42.04** | 31.80 | **633.85** | 4.53 | **375.23** | 8.86 | **81.55** | 35.01 |

# D  COMPLETE VERIFICATION WITH LiRPA ON CPU VS GPU

For a fair comparison, we only use one CPU core and one GPU (the same as GNN and GNN-ONLINE) in our experimental results in Section 4. In this section, we investigate the performance of our algorithm for the cases where one or multiple CPU cores are available without GPU acceleration. Note that existing baselines such as BABSR and MIPPLANET can only effectively utilize one CPU core subject to the Gurobi solver. GNN and GNN-ONLINE can utilize one GPU to run the GNN during branching while the rest of the verification processes all perform on one CPU core. In contrast, our method is much more flexible, and we are not limited by the number of CPU cores or GPUs. When running on multi-core CPUs, LiRPA can be automatically accelerated by the underlying linear algebra library (e.g., Intel MKL or OpenBLAS) since the main computation of LiRPA is just matrix multiplications.

In Figure 5, we show the performance of our algorithm on a single CPU core and multiple CPU cores (in blue), and compare it to our main results with one CPU core plus one GPU (in red). As we can see, the running time decreases when the number of CPU cores increases, but the speedup is not linear due to the limitation of the underlying linear algebra library and hardware. There is a big gap between the running time on 8 CPU cores and the time on one CPU core + one GPU, and the performance gap is more obvious on Wide and Deep models. Thus, the speedup of LiRPA computation on GPUs is significant. However, surprisingly, even when using only one CPU core, we are still significantly faster than baseline BABSR and also get very competitive performance when compared to GNN-ONLINE which needs one GPU additionally. This shows the efficiency of LiRPA based verification algorithms.

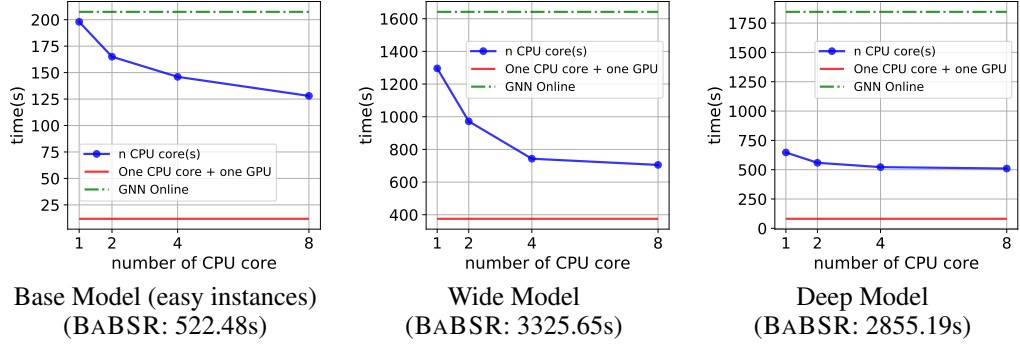

Base Model (easy instances)
(BABSR: 522.48s)

Wide Model
(BABSR: 3325.65s)

Deep Model
(BABSR: 2855.19s)

Figure 5: Running time of our method on the Base, Wide, and Deep networks when using 1, 2, 4 and 8 CPU cores without a GPU (blue), and our method using 1 CPU core + 1 GPU (red) and a strong baseline method, GNN-ONLINE (green). We report the baseline BABSR verification time in captions because they are out of range on the figures.

