# OpenReview forum: "Fast and Complete: Enabling Complete Neural Network Verification with Rapid and Massively Parallel Incomplete Verifiers"
_ICLR.cc/2021/Conference — ICLR 2021 Poster_

### Official Review · AnonReviewer1 · 2020-10-26
**Verifying simple neural network properties on a GPU**

**Rating:** 5
**Confidence:** 2

**Review:**

The paper focuses on verifying simple properties of neural networks on
accelerator hardware.  Instead of using linear programming, Lirpa is
considered as an alternative and minimal amounts of LP is added around
to allow to use the same class of properties. In the considered
examples the approach becomes much faster than the LP approach.

My main problem with the paper, is that is claims a complete
verification procedure without proper proofs. Yes, the new algorithm
is presented and some general discussion of things that are done with
it and how they work is provided. However, to claim complete
verification a general soundness of the procedure should be proved. In
particular I would like a theorem for the correctness of each of the
various components and a combined theorem for the whole procedure.

On the other hand, the experiments show that the approach is fast and
as such makes it more feasible for verification and the paper mostly
reads well.

An interesting alternative to discuss in related work, could be proof
assistants. See for example the work of Bentkamp, Blanchette JAR 2019.
Using proof assistants based on more complex logics, one can verify
properties much more efficiently.  The authors only mention Katz's
work on SMT, but if you consider higher order logic and the logics of
interactive theorem provers, the "NP-hard properties" can be checked
without considering all the cases.

Minor:
Conclusion starts with lowercase "we".

Based on the problems found by the other reviews and having read the rebuttals I have modified my score.

---

> ### Author Response · Authors · 2020-11-17
> **We have added theorems and the additional reference**
>
> We appreciate the helpful reviews from you, and we would like to address your concerns below:
>
> 1. Proofs:
>
> In our revised paper, we have greatly improved every section. In section 2, we discussed the soundness of LiRPA method. In section 3, we gave two Theorems to show the completeness of our full branch and bound algorithm based on the property of LiRPA method. In fact, the proof for correctness of branch and bound is relatively straightforward, and many previous works in complete verification [1] [2] [3] did not give an explicit proof and just imply it is correct (the proof would be very similar in every work). However, we have added proofs in our paper to ensure clarity. Specifically, in Section 3.2, we have discussed the completeness in Theorem 3.1 and 3.2. Theorem 3.1 shows that feasibility checking is important when using LiRPA as the bounding procedure in branch and bound, and Theorem 3.2 shows that with feasibility checking from LP, completeness is obtained, just like other works using branch and bound [1] [2] [3].
>
> 2. Additional reference:
>
> Thank you for providing this insightful connection! We have cited [Bentkamp, Blanchette JAR 2019] on proof assistant based higher order logic provers.
>
> Lastly, we hope the reviewer can check out our revised paper. We gave more clear formal definitions of the verification problem, as well as more intuition, background and examples. We also include detailed discussions on soundness and completeness as well as proofs for correctness. We hope the reviewer can re-evaluate our paper based on our revision and update the rating. Thank you.
>
> [1]Rudy Bunel, Jingyue Lu, Ilker Turkaslan, P Kohli, P Torr, and P Mudigonda. Branch and bound for piecewise linear neural network verification. Journal of Machine Learning Research, 21(2020)
>
> [2]Rudy R Bunel, Ilker Turkaslan, Philip Torr, Pushmeet Kohli, and Pawan K Mudigonda. A unified view of piecewise linear neural network verification. In Advances in Neural Information Processing Systems, pp. 4790–4799, 2018.
>
> [3] Jingyue Lu and M Pawan Kumar. Neural network branching for neural network verification. International Conference on Learning Representation (ICLR), 2020.

---

> ### Author Response · Authors · 2020-11-25
> **We made significant improvements on our paper and resolved problems found by other reviewers. We feel it is unjustifiable to decrease the rating.**
>
> Dear AnonReviewer1,
>
> Thank you so much for taking the time to read our response and comments from other reviewers.
>
> The most problems by other reviewers were on the writing and presentation of our paper (e.g., hard to understand, language issues, unclear definitions, etc). During the rebuttal period, we took this opportunity and greatly improved our paper. We have made a tremendous effort during the rebuttal period, not only to fix all problems reported by reviewers, but also significantly improved the quality of our paper. We have rewritten many paragraphs for enhancing clarity, included examples and nice-looking figures to explain our algorithm, and also reorganized and enriched all sections to ease understanding. We polished our paper several times to fix language issues.
>
> Additionally, we presented theorems and proofs as you and other reviewers suggested, and also cited the reference you mentioned. We believe we have addressed questions by all reviewers.
>
> Since the problems found by other reviewers were based on the initial submission of our paper, and we have significantly improved our paper during the rebuttal period, we hope the reviewer can reevaluate our paper based on our latest revision. We feel it is unjustifiable to decrease the rating of our paper despite our tremendous efforts on addressing problems from all reviewers and significantly improving our paper.
>
> We would like to thank you again for your constructive feedback. Could you please point out if there are any additional questions that we were not able to address in our responses? Thank you.
>
> Sincerely,
>
> Paper 3717 Authors

---

### Official Review · AnonReviewer4 · 2020-10-28
**Difficult to understand**

**Rating:** 5
**Confidence:** 1

**Review:**

### Summary
This paper describes a branch-and-bound (BaB) process for neural network verification that uses linear relaxation based perturbation analysis (LiRPA). It gives a way to tighten the bounds obtained via LiRPA. Overall, this results is a complete verification procedure, which is an order of magnitude faster than existing linear programming (LP) procedures.

### Strengths
The biggest strength of the paper is the impressive experimental results in section 5: the method described in the paper is several times faster than previous methods.

### Concerns
My main concern is that the paper is very difficult to understand. It seems to require a lot of background knowledge about the problem and the related literature, which is not clearly provided in the paper. I had trouble understanding the problem, the setup, and the proposed algorithm.

Another concern is that the paper claims that the proposed verifier is complete, but there is no proof of that. It does not seem like that's something too difficult to prove (given that BaB + LP is complete), but it should still be clearly stated.

Finally, the claim that the proposed framework outperforms previous methods by "at least 10X and up to 50X" is unsupported. Based on the results in section 5, a fairer statement regarding the speed would be "at least 3X and up to 15X" faster.

### Reasons for score
It is very difficult to judge this contribution, as the paper is hard to understand. The two main reasons for the score I give are 1) some of the claims of the paper are unsupported (see above) and 2) I believe this paper will have a much better chance of conveying the idea and making a contribution, if more background knowledge and intuition is provided throughout.

### Suggestions for improvement that have not affected the score I gave to the paper

One way to significantly improve the paper is to introduce more examples. An example consisting of a simple neural network to refer to throughout the explanation of LiRPA and BaB, and also in section 4, would make the paper much easier to read.

As a reader, I felt I couldn't appreciate the related work section so early in the paper. I encourage you to either move it later in the paper, or even better: introduce more background/examples in the introduction, as well as the notion of completeness, so that the related work is easier to understand.

Some typos:
* Abstract: "we demonstrate over a magnitude speedup ..." -> "we demonstrate speedup of an order of magnitude ..."
* First paragraph of section 2: "guarantee to terminate either ..." -> "guarantee to terminate when either ..."
* First paragraph of page 3: "used in state-of-the-art verifier (Lu & Kumar, 2020)" -> "used in the state-of-the-art verifier by Lu & Kumar (2020)".
* Second paragraph of page 3: "Our paper firstly leverage ..." -> "Our paper firstly leverages ..."
* First paragraph of section 3.1: "linear functions in the form of ..." -> "linear functions of the form ..."
* Start of section 4.1.: "As we have introduced ..." -> "As we discussed ..."
* Mid page 5: "greatly limited ..." -> "greatly limits ..."
* Bottom of page 5: "We follow the most challenge experimental setup ..." -> "We follow the most challenging experimental setup ..."
* Mid page 6: "we quickly reaches ..." -> "we quickly reach ..."
* Mid page 6: "with only two hidden node ..." -> "with only two hidden nodes ..."
* Section 4.3: "Benefited from our design ..." -> "Benefitting from our design ..."
* Conclusion: capitalisation of the first sentence

### Post rebuttal

Thank you to the authors for their detailed response and their effort in improving the presentation of the paper. I was impressed with how much the paper improved in this second version. In particular, I very much appreciate that the introduction starts with a simple one sentence explanation of the problem of neural network verification. This can be further improved if it included (almost) no maths, which can be deferred to the Background section. The figures in the updated paper are very good and a huge improvement of presentation. Finally, the paper now includes two clearly stated theorems, which also make the presentation and contribution much clearer.

I have increased the score I gave to the paper. Regardless of what the outcome for ICLR will be, I would like to encourage the authors to re-iterate on the presentation to really crystallize the problem, definitions and the suggested approach --- the paper is already so much better than the first version, and even just a little more work can make it even better.

---

> ### Author Response · Authors · 2020-11-17
> **Paper greatly improved, easier to understand. Added examples, inituitions, and proofs for completeness.**
>
> We really thank the reviewer for all the suggestions on improving our paper, and help us find many typos. We address your concerns below:
>
> ### Concern 1: Hard to understand, require a lot of background knowledge
>
> Thank you for pointing out this concern. We have greatly improved our paper in terms of introducing sufficient background and motivations especially for researchers outside of this field. In our revised Introduction, we have followed your suggestions and made the definition of verification problem very clear (in Introduction), and given a detailed walkthrough of background (section 2), and examples and figures to illustrate our main idea (Figure 1 and 2).
>
> ### Concern 2: Proof for completeness:
>
> We include more discussions on soundness and completeness in Section 2 and 3. In Section 2, the soundness of LiRPA has been proved in previous work (Xu et al., 2020) and we add a discussion on page 4. In Section 3.2, we have discussed the completeness in Theorem 3.1 and 3.2. Theorem 3.1 shows that feasibility checking is important when using LiRPA as the bounding procedure in branch and bound, and Theorem 3.2 shows that with feasibility checking from LP, completeness is obtained, just like other works using branch and bound [1] [2] [3].
>
> It is worth noting that many existing important works on complete verification such as [1] [2] [3] do not have a completeness proof, and it seems the completeness of the BaB process is implied, so most papers did not give proofs explicitly, and such a proof can look almost the same in every work. However, we do agree with the reviewer that we need to discuss more on the completeness of our algorithm and add explicit theorems for completeness, and we have done so in our revision.
>
> ### Concern 3: Speedup claim:
>
> Thank you for pointing this out! We forgot to update these numbers in introduction when our experiment results were updated. We have fixed the speedup claims in our paper. Our speedup is 30X compared to basic BaB baselines, and up to 5X compared to the state-of-the-art verifier. In our revision, we also added a very recent baseline (proximal BaBSR) and our method still performs best.
>
> ### Suggestion 1: Introduce more examples
>
> Following the reviewer’s suggestion, we have added Figure 2 to explain LiRPA bounds and the BaB procedure. Additionally, we also introduce more examples in the text: for example, in the introduction, we give the definition of the verification problem and show the example after the definition. We hope these updates will make our paper much easier to understand.
>
> ### Suggestion 2: Related work section
>
> As suggested by the reviewer, we have moved the related work section to the end of the paper, and enhanced the background section. We introduce the basic definitions of verification problems and the notation of completeness as early as in Introduction, and also give more former notations in section 2.
>
> ### Typos:
>
> Thank you for pointing out these typos. We have fixed them and also greatly improved writing and clarity in our new revision. We added the definition of verification, completeness and incompleteness in Introduction.
>
> ### Conclusion:
>
> We have significantly improved the writing of our paper and provide sufficient background and intuitions in our updated paper. We have also formally discussed and proven the completeness of our proposed algorithm. Since the most concerns on our paper are about writing and representation, we hope these changes address the concerns of the reviewer, and hope the reviewer can reconsider the score based on our revision. Feel free to let us know if you have any further questions regarding our revision. Thank you.
>
> [1]Rudy Bunel, Jingyue Lu, Ilker Turkaslan, P Kohli, P Torr, and P Mudigonda. Branch and bound for piecewise linear neural network verification. Journal of Machine Learning Research, 21(2020)
>
> [2]Rudy R Bunel, Ilker Turkaslan, Philip Torr, Pushmeet Kohli, and Pawan K Mudigonda. A unified view of piecewise linear neural network verification. In Advances in Neural Information Processing Systems, pp. 4790–4799, 2018.
>
> [3] Jingyue Lu and M Pawan Kumar. Neural network branching for neural network verification. International Conference on Learning Representation (ICLR), 2020.

---

> ### Author Response · Authors · 2020-11-22
> **Could you please reevaluate our paper based on our revision and response? Thank you!**
>
> Dear AnonReviewer4,
>
> We hope the reviewer can re-evaluate our paper based on our updated revision and detailed response, since the second stage of the discussion period is closing soon.
>
> As suggested by the reviewer, we have added detailed background on verification and branch and bound (Section 2), and also include formal proofs for the completeness of our methods (Theorem 3.1 and 3.2). We provide an example of LiRPA and BaB in Figure 2. We fixed typos, rewrote many paragraphs for easier understanding, and clarified our speedup claim. We moved the related work section to the last and also refined it as suggested.
>
> We believe the low rating from the reviewer is mostly based on the inadequate presentation of our idea and unpolished writing of the initial draft, but the technical contribution of our paper is solid, which is also recognized by the highly confident AnonReviewer2. We hope the reviewer can re-evaluate our paper based on the latest revision. We believe our paper is much easier to understand now.
>
> We really appreciate the very constructive review from the reviewer. Your suggestions have helped us improve our paper a lot. Please feel free to let us know any additional questions you may have. Thank you.
>
> Sincerely,
>
> Paper 3717 Authors

---

> > ### Comment · AnonReviewer4 · 2020-11-24
> > **Impressive improvement of presentation**
> >
> > Dear authors of paper 3717,
> >
> > Thank you for your detailed response and for your efforts in improving the paper. I would like to compliment you on a great improvement in presentation! I really like the figures and the more non-expert-friendly introduction. I've increased my score to 5 in light of this update.
> >
> > Regards,
> > AnonReviewer4

---

> > > ### Author Response · Authors · 2020-11-25
> > > **Thank you so much for reading through our revised paper and the encouraging comments**
> > >
> > > Dear AnonReviewer4,
> > >
> > > Thank you so much for reading through our revised paper! We are very grateful to you for your encouraging comments and we are happy to know that our paper is much better presented and more friendly to non-experts now!
> > >
> > > During the next stage of discussion, we hope you can also let other reviewers know that we have greatly improved our paper. We believe our technical contribution is clear and important, but the low ratings from AnonReviewer1 and AnonReviewer 3 were based on the bad first impression from our unpolished initial submission. We will really appreciate your support for our paper. Thank you!
> > >
> > > Sincerely,
> > >
> > > Paper 3717 Authors

---

### Official Review · AnonReviewer3 · 2020-10-28
**No Proof of Completeness?**

**Rating:** 5
**Confidence:** 2

**Review:**

The work proposes a new algorithm that can be used for the complete verification of neural networks (NNs). Unfortunately, the authors do not define the verification problem they study: Based on the second paragraph of the introduction, one is given a neural network (NN) on the input, and the task is to determine whether the NN has a specific formally defined property - but which kind of properties are verified is never explained. Intuitively, one would expect verification to focus on determining whether the NN gives a "correct" output for certain inputs, but that does not really match the general description given in the paper, and I did not find a place where the verification problem is formalized further. Without knowing what "verification" means in the context of this paper, it's difficult to follow the reasoning provided in the paper without having some rough idea of what kind of properties one wishes to verify (for instance, the discussion about using LP bounds assumes that the property that is being verified can be expressed using LP). I believe this issue could have been avoided by formalizing the precise task of verification (the verification problem).

On that note, many parts of the paper seemed rather confusing and hard to follow, either due to inconsistencies or due to language issues. For instance, the sentence "Input domain split is shown effective in verifying the properties with low input dimensions while performs as poorly as incomplete verifiers on higher dimension properties" on page 2 seems to contradict the definitions given for complete and incomplete verification. How can a complete verifier perform "as poorly as incomplete verifiers" if, by the definition given in the paper, complete verifiers must always correctly determine whether the NN has the given "property" or not? (Section 2: "Complete verifiers guarantee to terminate either the property is proved or a violation is located.")

In terms of presentation, the submission contains an incredibly large number of minor language issues (roughly 1 per 2-3 lines on average, ranging from minor article issues to malformed sentences; see also the quote in the previous paragraph), and I strongly encourage authors to fix these as they have a rather disruptive effect when trying to read and understand the paper. A very small number of examples is provided below:
Page 1
-"cause the changes of NN predictions" -> "cause changes of NN predictions"
-"Recently, a framework of Branch and Bound (BaB) (Bunel et al., 2018) is widely used for efficiently verifying NNs" - cannot combine "recently" and "is".
-"adopts Linear Program (LP) bounding procedure" -> "adopts a Linear Program (LP) bounding procedure"
Page 2
-"for construct LPs" -> "for constructing LPs"

The main contribution of the paper is the use of incomplete verifiers for complete verification, and the authors propose an algorithm for doing that using LIRPA bounds. However, I found no proof (or anything resembling a proof) showing that the resulting algorithm is correct, i.e., that it performs complete verification for neural networks. In fact, the problem is not even properly and formally defined in the paper. Hence, regardless of the experimental results, I do not think that the submission is ready for publication at this stage.

Post-Rebuttal Comment:
I thank the authors for responding to my comments. The updated version fixes most of the criticisms raised in the review, and I have raised the score accordingly. My new score is "5", partly because I believe that after performing such a large-scale and comprehensive overhaul of the paper (which was certainly necessary), the paper should go through a full new reviewing process.

---

> ### Author Response · Authors · 2020-11-17
> **Proof has been formally added; paper revised with a clear definition of problem**
>
> We greatly appreciate the very helpful comments from the reviewer. We have added necessary proofs, reorganized and revised many parts of our paper to make it easier to understand. We hope the reviewer can check out the updated version of our paper. We provide detailed answers below:
>
> 1. Formalization and definition of the task of verification:
>
> To make the verification problem very clear, we now have added a definition at the beginning of introduction, and also give an example after the definition. In section 2, we also give a more formal definition with discussions. We forgot to clearly define the properties under verification because the benchmarks used in our paper are fairly standard in this field, but now it has also been added everywhere, including in the experiment section. We hope it is now easier to understand the precise task of verification.
>
> 2. Proofs:
>
> We include more discussions on soundness and completeness in Section 2 and 3. In Section 2, the soundness of LiRPA has been proved in previous work (Xu et al., 2020) and we add a discussion on page 4. In Section 3.2, we have discussed the completeness in Theorem 3.1 and 3.2. Theorem 3.1 shows that feasibility checking is important when using LiRPA as the bounding procedure in branch and bound, and Theorem 3.2 shows that with feasibility checking from LP, completeness is obtained, just like other works using branch and bound [1] [2] [3].
>
> It is worth noting that many existing important works on complete verification such as [1] [2] [3] do not have a completeness proof, and it seems the completeness of the BaB process is implied, so most papers did not give proofs explicitly, and such a proof can look almost the same in every work. However, we do agree with the reviewer that we need to discuss more on the completeness of our algorithm and add explicit theorems for completeness, and we have done so in our revision.
>
> 3. Paper hard to follow, confusing sentences, language issues, typos
>
> We have fixed all confusing sentences and typos you mentioned and also greatly improve the writing and language of this paper. We added a clear definition of the verification problem, and also used more formal language to introduce the background and our algorithm in section 2 and 3. We also added figures and examples to illustrate our ideas more clearly. We rewrote those sentences that were hard to understand. The overall flow and clarity of our paper have greatly improved now, and we hope you can take a look again.
>
> Conclusions:
>
> We would like to thank the reviewer again for your constructive feedback. We hope you can read our revised paper once again where we have made great effort to improve writing and readability. We hope our answers address all your concerns and hope you can re-evaluate our revised paper, because the main issue was mainly language and representation problems rather than technical ones. Please kindly let us know if you have any additional comments. Thank you.
>
>
>
> [1]Rudy Bunel, Jingyue Lu, Ilker Turkaslan, P Kohli, P Torr, and P Mudigonda. Branch and bound for piecewise linear neural network verification. Journal of Machine Learning Research, 21(2020)
>
> [2] Jingyue Lu and M Pawan Kumar. Neural network branching for neural network verification. International Conference on Learning Representation (ICLR), 2020.
>
> [2]Rudy R Bunel, Ilker Turkaslan, Philip Torr, Pushmeet Kohli, and Pawan K Mudigonda. A unified view of piecewise linear neural network verification. In Advances in Neural Information Processing Systems, pp. 4790–4799, 2018.

---

> ### Author Response · Authors · 2020-11-22
> **Could you reevaluate our paper based on our revision and response? Thank you.**
>
> Dear AnonReviewer3,
>
> Since the second stage of the discussion period is closing soon, we hope the reviewer can re-evaluate our paper based on our updated paper and detailed response.
>
> In short, we have added a precise definition of the verification problem, and formal proofs for the completeness of our methods (Theorem 3.1 and 3.2). We also greatly improved the writing of our paper and fixed many language and consistency issues. Many paragraphs are rewritten for better clarity and the entire paper is polished several times.
>
> We believe the low rating from the reviewer is mostly based on the inadequate presentation of our idea and unpolished writing of the initial draft. The technical contribution of our paper is solid, which is also recognized by the highly confident AnonReviewer2. We hope the reviewer can re-evaluate our paper based on the latest revision. We believe our paper is much easier to follow now and has reached the quality for publication.
>
> We sincerely thank the reviewer again for the very insightful review, which greatly helped us to improve our paper. We will be glad to answer any additional questions you may have. Thank you.
>
> Sincerely,
>
> Paper 3717 Authors

---

### Official Review · AnonReviewer2 · 2020-10-29
**Strong experimental results, straightforward approach**

**Rating:** 7
**Confidence:** 4

**Review:**

The authors demonstrate that using a modification of the LiRPA method during the branch-and-bound process for solving the neural network verification problem can lead to significant speed-ups. The experimental results are strong. The authors convincingly show that the their method outperforms the existing state-of-the-art method by Lu & Kumar (2020) on an experimental setup similar to that work. The application of LiRPA to branch-and-bound is straightforward (since any incomplete verifier can be used), as is the use of gradient descent to improve the bound given by LiRPA (a standard technique applied to improve the bounds of certain verifiers).

Despite the fairly straightforward approach, the strength of the empirical results deserves attention. Overall, a solid contribution to the literature, and proof that research on incomplete verifiers leads to better complete verifiers.

Some questions/requests:

- The experimental setup details should be provided in the final version.

- How dependent is the performance of LiRPA on GPUs? For example, if we do a CPU-only comparison between the different methods, would other methods now outperform LiRPA? And if so by how much? What if we use multiple cores? I would understand if a detailed comparison is too computationally intensive, but I would like some sense of this.

---------------------------
Update after author response:
I thank the authors of the paper for significantly improving the prose of the paper, and I agree that the changes make the paper more self-contained and approachable. I have kept my ratings as my score was for primarily for the strong experiment results (and the score was also conditional on the paper being more polished). I am happy to support this paper for acceptance, but I am a little concerned about the degree of changes in the final version versus the initial submission, given the number of concerns the other reviewers had.

---

> ### Author Response · Authors · 2020-11-17
> **New experiments on LiRPA on GPU vs CPU, revised paper**
>
> Thank you so much for correctly recognizing the main contributions of our paper. We greatly appreciate your encouraging comments, and we are glad to answer your questions below.
>
> 1. Additional experimental setup: we have updated our paper and provide experimental setup details in Appendix B. We will release our full source code once accepted.
>
> 2. As suggested by the reviewer, we provided results on using single and multi-core CPU based LiRPA computation for our algorithm in Appendix C. Existing baselines such as BaBSR require an LP solver, which is hard to accelerate on GPU or even on multi-core CPU.  The basic computation of LiRPA is just matrix multiplication (like NN training), so it naturally enjoys the parallelization in existing deep learning software libraries such as Pytorch. Figure 5 in Appendix shows that our LiRPA based method on a single CPU core is still competitive when compared to BaBSR+LP on a single CPU, and we enjoy a speedup on multi-core CPUs.
>
> Additionally, we have greatly improved our paper in our revision, added examples to illustrate our problem under study and make the formulation and correctness of our algorithm more clear. We hope the reviewer can discuss our main contributions with other reviewers during the second and third stages of discussion. Thank you.

---

> ### Author Response · Authors · 2020-11-25
> **We have greatly improved the presentation and clarity of our paper, and hope the reviewer can champion our paper**
>
> Dear AnonReviewer2,
>
> We would like to thank you again for recognizing our main contributions and providing the very constructive reviews. We have added the additional experiments as you suggested, and also included an ablation study.
>
> Our initial submission was unpolished and was hard to understand for non-experts, and probably gave other reviewers a bad impression. However, we made a tremendous effort during the rebuttal period and significantly improved the quality of our paper. We have polished the paper for several rounds and greatly improved writing, clarity and presentation. We believe we have addressed the concerns on the readiness for publication.
>
> Because you are the only expert reviewer who are familiar with this field and understand our main contributions well, we hope you can champion our paper during the next stage of discussions and clarify our contributions to other reviewers who are not very confident. We believe our revised version is significantly better than our initial submission in terms of writing and clarity, while our contributions and main results remain unchanged. Thank you for your help and support!
>
> Sincerely,
>
> Authors of Paper 3717

---

### Public Comment · ~Alessandro_De_Palma1 · 2020-11-15
**Interesting work that lacks a detailed presentation and analysis**

I believe this is an interesting work and contains some valuable ideas for the neural network verification community, such as the joint optimization on intermediate bounds and output bounds. However, the paper is lacking crucial details and inaccurately describes the state of the art for complete neural network verification.

For instance, in the introduction, it is claimed that LiRPA-based bounds “have never been explored as the main driver in the complete verification settings”. However, LiRPA bounding has been successfully employed in the past as last layer bounding for complete verification. For instance, by [1] (see Bunel et al., (2020b), section 4.1). Moreover, GPU-accelerated and massively parallel branch and bound frameworks already exist. The work by Bunel et al. (2020a) does not rely on LP solvers and, similarly to the proposed approach, iteratively tightens the bounds of a LiRPA method (Wong and Kolter, 2018), from which it is initialized.

The main contribution of the paper is then a gradient-descent based approach that optimizes over the lower bound slopes of LiRPA-based bounds.
The choice of the LiRPA formulation seems to allow for joint optimization over intermediate bounds, and I believe this is the key to the works’ empirical success.
I would like to ask the authors for the following clarifications:

1)	Let us assume that intermediate bounds are kept fixed throughout the procedure. Which bounds will this optimization converge to? Due to the primal-dual correspondence of LiRPA methods described by Salman et al. (2019), it seems likely that the authors might be solving a formulation resembling the dual presented in Theorem 1 of (Wong and Kolter, 2018).
2)	How is the tightening of intermediate bounds performed?  Are the intermediate slopes ($D^i_l$) obtained from the output bounding simply plugged in the LiRPA problems for the intermediate bounds? If so, is this done after each iteration of gradient descent?
3)	In section 4.1, the authors seem to suggest that the upper intermediate slopes ($D^i_u$) are treated as a function of the lower slopes ($D^i_l$) before them. Does this make the optimization problem non-convex? If so, how is the non-convexity handled?

Additionally, I believe the empirical comparison would benefit from the addition of a GPU-accelerated baseline such as the work by Bunel et al., (2020a), whose complete verification pipeline is available at [2]. Moreover, an ablation study measuring the effect of the tightened intermediate bounds would be quite interesting.

[1] Shiqi Wang, Kexin Pei, Justin Whitehouse, Junfeng Yang, and Suman Jana. Formal security analysis of neural networks using symbolic intervals. USENIX 2018.
[2] https://github.com/verivital/vnn-comp/tree/master/2020/CNN/oval_framework

---

> ### Author Response · Authors · 2020-11-17
> **We made our statements more clear, added ablation study and comparisons to your work**
>
> Dear Alessandro,
>
> Thank you for your interests in our work and we really appreciate your valuable comments and insightful questions. In our revised version, we have greatly improved the clarity of our paper, and we hope you can check it again. We now respond to your individual comments below:
>
> ### 1. Usage of LiRPA bounding in previous works:
>
> Thank you for pointing this out. We have revised that sentence in Section 1 to make it more clear. Previous works using LiRPA bounding did not use the optimization procedure as we proposed in Section 3.1, so they only produce relatively loose bounds. As you mentioned, the success of our work can be attributed to the much tighter bound after a joint optimization of intermediate layer bound and final bound. We have added an ablation study to show the importance of the optimization process in Appendix C.
>
> Additionally [1] only conducts split on input domain to use the LiRPA bounds. In our paper, we split ReLU nodes instead, as it is usually more effective with high dimensional inputs (Bunel et al., 2018). In Theorem 3.1 we show that when splitting ReLU nodes, if LiRPA is used as the only bounding procedure, BaB is still incomplete. This is a critical observation that is also not discussed in previous works.
>
> ### 2. About your work (Bunel et al. 2020a):
>
> Thanks for clarifying the details of your great work (Bunel et al. 2020a). We have appreciated its contributions in terms of GPU acceleration without LP in Section 5 (related work) and we have provided detailed comparisons against it in Section 4 Table 1 and 2.
>
> Thank you so much for providing the github link. The code link included in paper (Bunel et al. 2020a) does not contain the complete verification part, so it was difficult for us to make a comparison. In our paper, we currently directly report the best numbers in (Bunel et al. 2020a) in our tables. Compared to that number, we are still overall (up to 5X) faster.
>
> We will look into the code you provided, make necessary adjustments, and reproduce the experiments in your paper. We plan to update the numbers for (Bunel et al. 2020a) in a later revision (those experiments can take some time).
>
> ### Questions 1. Which bounds will optimization converge to if intermediate bounds are fixed?
>
> If the intermediate bounds are fixed, the optimization converges to the optimal dual solution or the LP solution built on the same intermediate bounds. As you mentioned, this relationship has been shown in (Salman et al., 2019). However, the main difference here is that we also optimize intermediate bounds so stronger results can be obtained.
>
>
> ### Question 2. How is the tightening of intermediate bounds performed?
>
> Suppose we set the slope of lower bounds for all ReLU neurons as a vector variable, $\alpha$. Based on this slope, we can compute intermediate bounds as a function of $\alpha$. The final bound is a function of intermediate bounds and $\alpha$, so it is essentially just a complex function of $\alpha$. Then we use a gradient over $\alpha$ to optimize the final layer bound as the objective.
>
>
> The tightening procedure is actually very straightforward because we use auto_LiRPA, an automatically differentiable implementation of LiRPA. We do not need to manually derive this gradient, and it is actually a very complicated function because of interactions between the slopes in different layers (the bounds are computed recursively).
>
>
> ### Question 3. Non-convexity:
>
> Yes, the optimization problem is non-convex and significantly more complicated than LPs. However, any setting of a valid slope in [0, 1] yields a valid bound. So the non-convexity does not affect the soundness of our verifier. We use gradient based optimizers like Adam to optimize this bound. Convergence guarantee is not necessary here because it does not affect soundness. We do not aim to achieve the global optima, and in practice we actually only run a few steps (like 10 steps) of gradient descent which can greatly improve the solution in a very short time. The non-convexity is potentially helping us here, because it is a much more complicated objective than the previous LP dual based approach and can potentially yield tighter bounds. In practice, this non-convex problem can be sufficiently optimized with gradient descent and works very well.
>
>
> ### Question 4: Ablation study:
>
> We provided ablation study in Appendix C. We find that our optimized LiRPA bound is indeed the most important contributor here, while batch split also helps to further speedup.
>
>
> Lastly, We thank you again for your comments and we also enjoy your paper very much. We hope you can check the latest version of our paper and we would like to further discuss with you. We are still working on revising this paper, so feel free to let us know if you have any additional comments. Thank you.

---

### Author Response · Authors · 2020-11-24
**General Response: we greatly improved writing and presentation of our paper, and added proofs for completeness**

Dear Reviewers,

We really appreciate your constructive comments which are very helpful for improving our paper. During the discussion period, we take the opportunity to revise our paper based on the suggestions from reviewers. **We have significantly improved the writing and clarity of our paper**.

Specifically, we have significantly enhanced Section 2 for a comprehensive discussion of backgrounds, added a precise definition of the verification problem, included more examples with figures, and presented theorems for completeness. We rewrote many paragraphs for better clarity and also polished the paper several times to fix many language and consistency issues. Additionally, we also added ablation study experiments, comparisons to latest works and results on multi-core CPUs as requested by the reviewers. We believe our paper is much easier to follow now.

One question raised by the reviewers is that we did not state a theorem for the completeness of our algorithm. In our revision, we formally **added Theorem 3.1 and 3.2 and also included more discussions on the soundness and completeness** in Section 2.2, 2,3 and 3.2. We note that in many existing works on complete verification [1][2][3], the theorem for the completeness of branch and bound is omitted and not given explicitly (as they are conceptually straightforward), but we agree with the reviewers adding formal statements like our Theorem 3.1 and 3.2 is very helpful for understanding.

We feel the most concerns from the reviewers are on the clarity and presentation problems of our paper, and **our technical contributions are solid and impressive**, which is recognized by the highly confident AnonReviewer2. We hope our latest revision can address the concerns on the readiness for publication of our paper.

Since today is the last day of the discussion period, we will really appreciate it if the reviewers can go over our latest paper revision and our detailed response, and reevaluate our paper based on them. Please feel free to ask us any questions you may still have and we will be more than happy to answer them before the deadline (Nov 24 AoE).

Thank you again for reviewing our paper and we look forward to discussing with you.

Sincerely,

Paper 3717 Authors

References:

[1]Rudy Bunel, Jingyue Lu, Ilker Turkaslan, P Kohli, P Torr, and P Mudigonda. Branch and bound for piecewise linear neural network verification. Journal of Machine Learning Research, 21(2020)

[2]Rudy R Bunel, Ilker Turkaslan, Philip Torr, Pushmeet Kohli, and Pawan K Mudigonda. A unified view of piecewise linear neural network verification. In Advances in Neural Information Processing Systems, pp. 4790–4799, 2018.

[3] Jingyue Lu and M Pawan Kumar. Neural network branching for neural network verification. International Conference on Learning Representation (ICLR), 2020.

---

### Decision · Program_Chairs · 2021-01-07
**Final Decision**

**Decision:**

Accept (Poster)

**Comment:**

Thank you for your submission to ICLR.  As noted, several of the reviewers had fairly low confidence in evaluating this submission.  However, based upon the reviewers and commenters who were familiar with this line of work, as well as my own evaluation of the paper, I believe it is clearly worth publishing at ICLR.  The proposed method pushes the boundary in methods for exact branch and bound-based verification of neural networks, using clever tricks from existing relaxations.  And while the method is still likely to be relegated to relatively small networks for the time being, pushing forward the state of the art in exact verification is still a worthy goal suitable for publication at ICLR.  I thus think that the paper is quite clearly above the bar, and should be accepted for publication.